# MetaMind: Modeling Human Social Thoughts with Metacognitive Multi-Agent Systems

**Xuanming Zhang**[1], **Yuxuan Chen**[2], **Samuel Yeh**[1], **Sharon Li**[1]*
[1]Uniersity of Wisconsin-Madison
[2]Tsinghua University
xzhang2846@wisc.edu, sharonli@cs.wisc.edu

## Abstract

Human social interactions depend on the ability to infer others' unspoken intentions, emotions, and beliefs—a cognitive skill grounded in the psychological concept of Theory of Mind (ToM). While large language models (LLMs) excel in semantic understanding tasks, they struggle with the ambiguity and contextual nuance inherent in human communication. To bridge this gap, we introduce **MetaMind**, a multi-agent framework inspired by psychological theories of metacognition, designed to emulate human-like social reasoning. MetaMind decomposes social understanding into three collaborative stages: (1) a *Theory-of-Mind Agent* generates hypotheses about user mental states (e.g., intent, emotion), (2) a *Moral Agent* refines these hypotheses using cultural norms and ethical constraints, and (3) a *Response Agent* generates contextually appropriate responses while validating alignment with inferred intent. Our framework achieves state-of-the-art performance across three challenging benchmarks, with 35.7% improvement in real-world social scenarios and 6.2% gain in ToM reasoning. Notably, it enables LLMs to match human-level performance on key ToM tasks for the first time. Ablation studies confirm the necessity of all components, which showcase the framework's ability to balance contextual plausibility, social appropriateness, and user adaptation. This work advances AI systems toward human-like social intelligence, with applications in empathetic dialogue and culturally sensitive interactions. Code is available at https://github.com/XMZhangAI/MetaMind.

## 1 Introduction

> *"What is meant often goes far beyond what is said, and that is what makes conversation possible."*
>
> — H. P. GRICE

Everyday human conversation can be filled with intent that goes unspoken—feelings implied but never named, expectations hinted at with no explicit instruction, and suggestions masked as statements. Consider the utterance: *"It's cold in here."* Depending on who says it, to whom, and in what context, it could be a mere observation, a polite request to close a window, or even an expression of discomfort seeking empathy. Humans handle such ambiguity by reasoning about the speaker's beliefs, desires, emotions, thoughts, and intentions—mental states that are not directly observable. This capacity, known as *Theory of Mind* (ToM) [1], has been extensively studied in the field of developmental psychology and is shown to emerge in children around the age of four [2, 3]. This allows humans to move beyond the literal surface of language and grasp the deeper intent behind what is said.

---

*Corresponding author.

39th Conference on Neural Information Processing Systems (NeurIPS 2025).

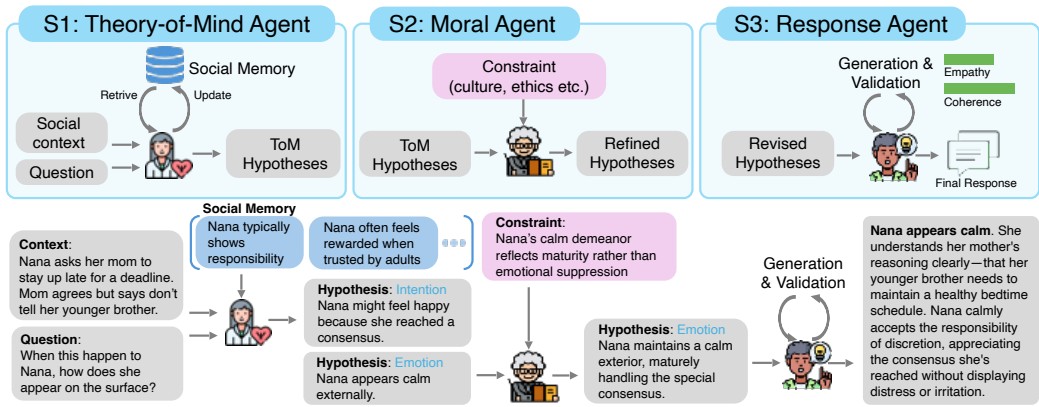

Figure 1: **MetaMind** multi-agent framework. The architecture comprises three collaborative agents—Theory-of-Mind Agent, Moral Agent, and Response Agent—working in a staged metacognitive loop. The ToM Agent generates hypotheses about latent mental states, which are refined by the Moral Agent using cultural/ethical constraints. The Response Agent synthesizes contextually appropriate outputs while validating them with inferred intent.

Large language models (LLMs), by contrast, often falter in this regard. While they excel in semantic understanding tasks by producing fluent and contextually relevant text [4], they struggle with social reasoning with ambiguity and indirectness that characterize real-world communication. For instance, LLMs can fall short in applications that demand human-like social intelligence, including empathetic dialogue and conflict mediation [5, 6]. Addressing this gap is essential for building AI systems that interact effectively in socially complex environments.

One of the key challenges in bridging this gap lies in inferring user mental states—beliefs, desires, emotions, and intentions—that are not directly observable but are essential for interpreting socially nuanced language. Unlike humans, LLMs do not naturally infer these unspoken intentions, making it particularly difficult for them to respond appropriately in scenarios involving indirect speech, implied emotions, or culturally sensitive cues [7–9]. Recent work has attempted to address these challenges by injecting social behavior into LLMs [10–12], such as simulating social interactions via static role-play prompting [13] or fine-tuning with preference data [14, 15]. However, these approaches largely optimize for surface-level statistical alignment and fail to capture the structured, multi-stage cognitive process humans use to reason about unobservable intent [9] and generalize across diverse cultural and social contexts [16, 17]. Most notably, they treat social reasoning as a single-step prediction problem, rather than a layered process involving interpretation, reflection, and adaptation—a hallmark of human metacognition [18, 1]. We argue that enabling LLMs with such staged reasoning capabilities is critical for achieving socially intelligent AI.

In this paper, we propose **MetaMind**, a cognitively motivated framework designed to explicitly model the key components in human-like social reasoning through a staged and collaborative multi-agent system. Our approach is grounded in psychological theories of metacognition [18, 1], which describe how humans reflect on their own thinking, revise their understanding in light of social norm constraints, and adapt their behavior in socially complex environments. MetaMind mirrors this layered reasoning process through three specialized agents, each responsible for a distinct stage of cognitive-social inference. ❶ A *Theory-of-Mind Agent* initiates reasoning by generating multiple hypotheses about the user's mental state based on contextual and social cues. This reflects the first step in human ToM: inferring what the speaker might be trying to convey beyond literal words. For example, when a user remarks that "work has been exhausting lately", the system may infer underlying burnout, frustration, or a need for empathy. ❷ A *Moral Agent* then revises and filters these candidate hypotheses by incorporating socially grounded constraints, such as cultural expectations, ethical norms, or situational appropriateness. Just as humans refine their initial interpretations by aligning with social context, this agent ensures that the model's reasoning remains socially responsible and context-aware. For instance, if romantic intent is hypothesized in a workplace conversation, the Moral Agent may reinterpret it as collegial admiration based on professional norms. ❸ Finally, a *Response Agent* generates and self-validates the output, conditioning on the refined optimal hypothesis and the user's social memory (*e.g.*, emotional patterns and prior preferences). This final step enacts

a metacognitive loop that allows the system to respond with greater empathy, nuance, and cultural sensitivity.

We conduct a comprehensive empirical evaluation of MetaMind across a suite of challenging social intelligence benchmarks, including ToM reasoning [19], social cognition, and social simulation [20] tasks. Our study spans over 16 contemporary LLMs, assessing both general social reasoning ability and performance in real-world, context-sensitive scenarios. Empirical results show that MetaMind achieves a **35.7%** average improvement on real social scenario tasks and a **9.0%** average gain in overall social cognition ability—substantially enhancing the social competence of underlying LLMs. Notably, our framework enables representative LLMs to *match average human performance on key benchmarks*. We also perform detailed ablation studies to isolate the contribution of each agent in the system, revealing that all three stages are critical to the framework's success.

We summarize our key contributions below:

- We propose MetaMind, a cognitively grounded, multi-agent framework that models human-like social reasoning by inferring mental states and incorporating social and ethical constraints, while adapting to user-specific patterns.
- We conduct comprehensive evaluations to demonstrate that MetaMind significantly improves both contextual accuracy and social appropriateness in real-world scenarios, achieving *state-of-the-art results* on challenging benchmarks and even matching human performance.
- We perform in-depth ablations to understand the impact of each agent and various design choices of our framework, justifying that all three components are essential to performance and generalization.

## 2 Related Work

**Theory of Mind in AI.** Prior work has explored simulating ToM in AI systems through diagnostic frameworks [7, 11, 10], revealing limitations of LLMs in inferring beliefs, intentions, and social nuances [8, 21, 22, 5, 9, 23]. While these studies identify key gaps—such as failures in handling recursive mental states or contextual ambiguities—their solutions often focus on narrow task-specific interventions, such as fine-tuning and testing on curated datasets [24, 6, 25, 26, 12, 13] or rule-based intent classifiers [27–30]. In contrast, we introduce a holistic framework grounded in theories of metacognition [18], which treats ToM not as a specialized task but as a foundational reasoning capability. Our framework integrates mental state inference, social norm constraints, and evolving social memory into a unified system, enabling generalized and context-sensitive social reasoning.

**Prompting and Parameterized Reasoning.** Methods like chain-of-thought prompting [31] and constrained decoding [32] aim to enhance LLMs' reasoning by structuring intermediate steps or injecting task-specific rules. However, these approaches lack mechanisms for contextual adaptation [9]. Similarly, role-play prompting [13] simulates social interactions but relies on static personas, failing to capture the fluid interplay of social intent and context-dependent rules. Alignment approaches like RLHF [14] and instruction tuning [15] have improved adherence to user intent, but scaling these methods poses challenges in data curation and generalization control. Our framework departs from existing paradigms by decomposing reasoning across *collaborative agents*, enabling multi-stage, self-reflective social reasoning skin to human metacognition.

**Multi-Agent LLM Systems.** Multi-agent LLM systems have been used across a wide range of tasks, including debate-style reasoning [33, 25, 34, 35], retrieval-augmented generation [36–38], and collaborative tool use [32, 39–41]. These systems typically assign agents specialized roles to divide and coordinate subtasks. However, the use of multi-agent frameworks for socially grounded reasoning remains relatively underexplored. While some studies have applied agent collaboration to simulate social interactions or role-play conversations [6, 13, 28, 42–47], they often focus on persona consistency, without modeling how agents can collaboratively infer and revise social interpretations. Our work addresses this gap by developing a multi-agent architecture specifically designed for social reasoning, in which agents interact not only to complete tasks but to interpret user mental state and incorporate social norms—mirroring core elements of human cognition.

**Metacognitive Architectures.** Psychological theories of metacognition posit that self-regulated learning and reasoning rely on iterative cycles of planning, self-monitoring, and evaluative reflection [18, 48]. While these principles are well-established in human cognition, their systematic

integration into LLM architectures remains underexplored [49, 50]. Current LLM-based systems often adopt oversimplified approaches, relying on monolithic prompting or partitioning functionality into isolated modules [51]. Our framework addresses this gap by formalizing metacognitive principles [52] into specialized, collaborative agents. This design mirrors human self-regulation, where adaptive reasoning emerges from synergistic interaction rather than static components.

# 3 Methodology

## 3.1 Stage 1: Generating Mental State Hypothesis via Theory-of-Mind (ToM) Agent

A core feature of human social cognition is the ability to attribute *unobservable* mental states—such as beliefs, desires, intentions, and emotions—a capacity broadly referred to as *Theory of Mind* [53, 3, 1]. This ability underpins what developmental psychologists call "folk psychology": our everyday, intuitive reasoning about how and why others act [54]. It enables us to infer latent intent behind indirect speech and interpret emotionally charged or ambiguous behavior. While LLMs excel at semantic reasoning, they struggle with ToM-driven reasoning, often defaulting to literal interpretations that miss latent user intent.

To address this gap, we introduce a dedicated ToM Agent that serves as the entry point in our metacognitive reasoning pipeline. Our design is grounded in theories of metacognition [18], where mental state attribution is treated as a structured inference process [55]. Rather than attempting to respond directly to user inputs, the ToM Agent seeks to construct a set of plausible interpretations of what the user might be thinking or feeling. The ToM Agent formalizes the process of mental state inference as *hypothesis generation*—grounded in context, social knowledge, and prior interactions—which will be refined and leveraged in subsequent stages.

**Hypothesis Generation.** Formally, given a user prompt $u_t$, the ToM Agent operates under a contextual input $\mathcal{X} = (u_t, C_t, M_t)$, where $C_t$ denotes the social context (i.e., previous conversational history), and $M_t$ denotes the social memory, which is a dynamic database storing user preferences and salient emotional markers (see details in Appendix A.4). Provided with the contextual input $\mathcal{X}$, the goal of the ToM Agent is to generate a set of candidate mental state interpretations $\mathcal{H}_t = \{h_1, h_2, \ldots, h_k\}$, where each $h_i \in \mathcal{Y}$ is an instantiation of a latent mental state, accompanied by natural language explanations and type labels from the set $\mathcal{T} = \{\texttt{Belief}, \texttt{Desire}, \texttt{Intention}, \texttt{Emotion}, \texttt{Thought}\}$.

The inference mechanism of the ToM Agent is implemented via *Mental-State Reasoning*. This procedure unfolds in four conceptual steps: (1) generating commonsense-based hypotheses from the input $(u_t, C_t)$, (2) cross-referencing these hypotheses with the social memory $M_t$, (3) identifying Theory-of-Mind markers across predefined categories, and (4) generating a set of $k$ candidate hypotheses belonging to the identified ToM marker. This structured reasoning encourages the model to simulate human-like inference processes by incorporating contextual grounding and hypothesis diversification. To instantiate this reasoning process, we define the prompt in Table A.1, which guides the language model to reason about the user question in a manner consistent with the psychological definition of Theory of Mind—namely, as an inferential process that constructs internal representations of others' minds using contextual and background knowledge. This explicit hypothesis generation stage enables subsequent modules to reason over a diverse set of plausible interpretations, rather than committing prematurely to a singular semantic response.

## 3.2 Stage 2: Refining Hypothesis via Moral Agent

The Moral Agent forms the second stage of our social reasoning pipeline and serves to *refine the hypothesis* generated by the ToM Agent. While the first stage focuses on what the user might be thinking or feeling, the Moral Agent assesses whether these interpretations are appropriate given broader norms—such as cultural norms and ethical constraints. This step ensures that the system not only understands intent, but also responds in a socially responsible and domain-aware manner.

**Hypothesis Refinement and Selection.** Formally, the Moral Agent takes as input the set of latent mental state hypotheses $\mathcal{H}_t = \{h_1, \ldots, h_k\}$ produced by the ToM Agent, along with a set of constraint rules $\mathcal{D}$. Each rule in $\mathcal{D}$ describes a specific norm or guideline, such as "Romantic suggestions are not appropriate in professional settings". These rules are encoded as conditions that

determine whether a hypothesis should be retained, reweighted, or revised. For instance, if the ToM Agent infers a romantic intention in a professional conversation, the role-based prompt will instruct the model to reinterpret this intent in a more appropriate way (*e.g.*, as a joke or misunderstanding).

The moral agent proceeds in two steps. First, for each original hypothesis $h_i \in \mathcal{H}_t$, the Moral Agent generates a revised version $\tilde{h}_i$ that incorporates the relevant domain rules. This revision may involve rephrasing the interpretation and adjusting its social tone. The revision process is implemented using targeted prompts instantiated for three types of rules: cultural norms, ethical constraints, and role-based expectations. We provide the prompt details in Appendix A.2.

Next, the agent selects the most appropriate revised hypothesis $\tilde{h}^*$ by scoring each candidate $\tilde{h}_i$ based on a composite objective:

$$\tilde{h}^* = \arg\max_i \; [\underbrace{\lambda \cdot P(\tilde{h}_i|u_t, C_t, M_t)}_{\text{Contextual plausibility}} + \underbrace{(1-\lambda) \cdot \log \frac{P(\tilde{h}_i|u_t, C_t, M_t)}{P(\tilde{h}_i)}}_{\text{Information Gain}}], \tag{1}$$

where the first term denotes the contextual plausibility of the revised hypothesis, and the second term reflects the implicit information gain of the revised hypothesis when considering the context. The weight $\lambda$ balances contextual plausibility and social appropriateness with how much the hypothesis is informed by the context versus being generic. The selected hypothesis $\tilde{h}^*$ is passed to the final stage for response generation and further validation.

### 3.3 Stage 3: Generating and Validating Output via Response Agent

The final stage of the system is responsible for generating a contextually appropriate response and validating its alignment with the inferred user intent. While the earlier stages focus on understanding and refining mental state interpretations, the Response Agent is tasked with transforming this structured understanding into a concrete action—typically a natural language response—while preserving coherence, empathy, and domain compliance.

**Generation and Validation.** This stage receives as input the final selected hypothesis $\tilde{h}^*$. Alongside, to ensure consistency with long-term user preferences and prior emotional states, the Response Agent incorporates social memory during decoding, enabling the model to adapt the tone or emotional framing of its response. The response $o_t = (y_1, y_2, \ldots, y_L)$ is generated by a decoder LLM as the follows:

$$o_t = \arg\max \prod_{t=1}^{L} p(y_t \mid y_{<t}, \tilde{h}^*, M_t, u_t),$$

which maximizes the likelihood of the response conditioned on the optimal interpretation and social memory. To ensure alignment between the generated response and the intended user state, the Response Agent includes a self-reflection mechanism, assessing its social and semantic quality using a utility score:

$$U(o_t) = \underbrace{\beta \cdot \text{Empathy}(o_t, u_t, M_t)}_{\text{Emotional alignment}} + \underbrace{(1-\beta) \cdot \text{Coherence}(o_t, C_t, \tilde{h}^*)}_{\text{Contextual coherence}}, \tag{2}$$

where $\text{Empathy}(\cdot)$ quantifies how well the response resonates with the user's inferred emotional or cognitive state, and $\text{Coherence}(\cdot)$ evaluates consistency with the conversational context and task constraints. The system can trigger regeneration if the utility score is too low. We provide the prompt details for both generation and validation in Appendix A.3.

## 4 Experiments

We evaluate MetaMind using three challenging benchmarks spanning Theory-of-Mind reasoning, social cognition, and social simulation tasks. Each benchmark naturally emphasizes a different aspect of the reasoning pipeline—aligning well with the core functionality of each stage in our multi-agent framework. In particular, ToMBench [19] aligns with the function of Theory-of-Mind Agent (Stage 1) by assessing the model's ability to infer latent mental states. Then, we employ a suite of social cognition tasks, which align with the Moral Agent's capacity in refining interpretations under

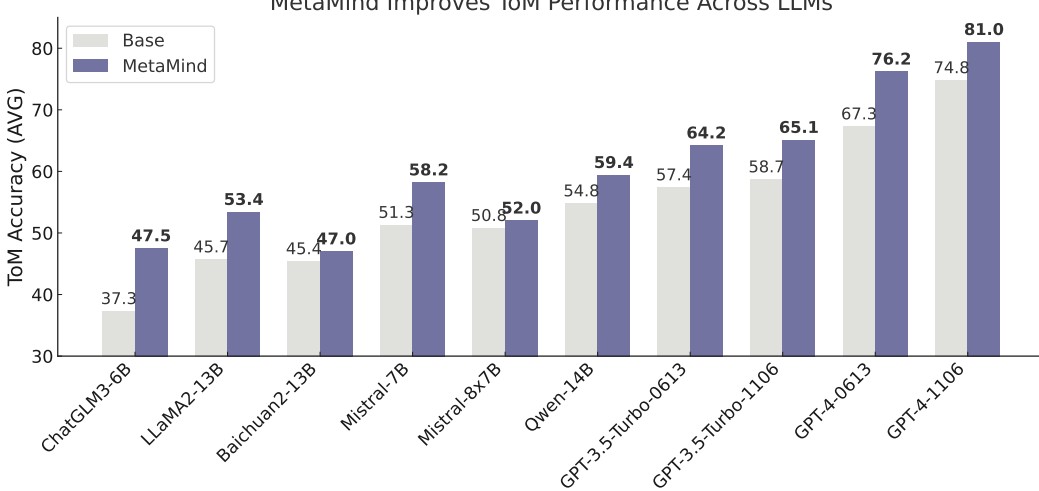

Figure 2: **MetaMind improves Theory-of-Mind reasoning performance across LLMs.** Each pair compares base model accuracy (gray) with MetaMind-enhanced accuracy (purple) on ToMBench. MetaMind consistently boosts ToM reasoning across both open-source and proprietary LLMs, highlighting its generality and effectiveness. See detailed performance in Appendix B.1.

normative and ethical constraints (Stage 2). Lastly, the STSS benchmark [20] focuses on open-ended, interactive scenarios that test the Response Agent's ability to generate contextually appropriate responses (Stage 3). Together, these benchmarks offer a comprehensive evaluation of MetaMind. For reproducibility, *we include implementation details and sensitivity analysis on hyperparameters (including $k$, $\lambda$, and $\beta$) in Appendix A.5*.

### 4.1 Theory-of-Mind Reasoning Task

We first evaluate MetaMind's ability to infer latent mental states using ToMBench [19], a multiple-choice benchmark designed to test Theory-of-Mind reasoning across six categories: Emotion, Desire, Intention, Knowledge, Belief, and Natural Language Communication. This benchmark aligns directly with the function of the Theory-of-Mind Agent in Stage 1, which is responsible for generating structured mental state hypotheses from indirect or ambiguous input.

MetaMind achieves new state-of-the-art performance on ability-oriented Theory-of-Mind reasoning, outperforming both base GPT-4 and competitive prompting-based baselines such as Chain-of-Thought [31], SymbolicToM [56]. As shown in Table 1, MetaMind-enhanced GPT-4 reaches an average accuracy from 74.8% to **81.0%**, surpassing all prior methods across most ToM dimensions. Importantly, MetaMind's improvements are not limited to GPT-4: additional experiments (Figure 2) show consistent performance gains *across diverse LLM backbones*, including open-source models like Mistral and Qwen. These results highlight the generality of our multi-agent framework and its effectiveness as a model-agnostic enhancement for social reasoning. For more information about baseline, see Appendix A.7.

Table 1: Comparison on Theory-of-Mind reasoning task.

|  | Emotion | Desire | Intention | Knowledge | Belief | NL Comm. | AVG. |
|---|---|---|---|---|---|---|---|
| Base (GPT-4) | 75.7 | 69.7 | **84.7** | 52.1 | 82.8 | 84.0 | 74.8 |
| w. CoT [31] | 73.2 | 63.3 | 77.9 | 60.4 | 83.6 | 83.0 | 73.6 |
| w. HM [57] | 76.4 | 71.1 | 80.2 | 59.3 | 84.1 | 85.0 | 76.0 |
| w. ToM2C [58] | 77.2 | 70.4 | 81.5 | 57.8 | 85.3 | 84.6 | 76.1 |
| w. Generative Agents [28] | 74.8 | 72.0 | 78.9 | 55.6 | 83.2 | 86.4 | 75.1 |
| w. SymbolicToM [56] | 75.9 | 70.9 | 79.6 | 58.2 | 84.0 | 83.7 | 75.4 |
| w. **MetaMind** (ours) | **78.7** | **76.5** | 84.3 | **68.2** | **88.6** | **88.5** | **81.0** |

Table 2: Comparison on social cognition tasks.

**UOT**: Unexpected Outcome Test  **SIT**: Scalar Implicature Task  **PST**: Persuasion Story Task  **FBT**: False Belief Task
**AST**: Ambiguous Story Task     **HT**: Hinting Test    **SST**: Strange Story Task     **FRT**: Faux-pas Recognition Test

| | UOT | SIT | PST | FBT | AST | HT | SST | FRT | AVG. |
|---|---|---|---|---|---|---|---|---|---|
| Base (GPT-4) | 71.0 | 49.0 | **65.0** | 88.2 | 77.5 | 82.5 | 84.0 | 73.3 | 71.5 |
| w. CoT [31] | 72.7 | 55.0 | 55.0 | 86.8 | 81.0 | 82.5 | 84.3 | 75.2 | 74.1 |
| w. HM [57] | 74.0 | 54.6 | 59.2 | 87.6 | 82.2 | 83.1 | 85.0 | 76.0 | 75.2 |
| w. ToM2C [58] | 75.3 | 52.9 | 60.4 | 88.0 | 80.1 | 84.4 | 83.7 | 77.8 | 75.3 |
| w. Generative Agents [28] | 73.2 | 56.8 | 57.8 | 87.1 | 83.6 | 81.9 | 85.8 | 75.9 | 75.3 |
| w. SymbolicToM [56] | 72.4 | 58.1 | 58.7 | 87.9 | 82.7 | 82.8 | 84.2 | 76.3 | 75.4 |
| **w. MetaMind (ours)** | **81.5** | **60.4** | 64.8 | **90.1** | **88.8** | **86.2** | **88.4** | **83.9** | **80.5** |

Table 3: Social simulation performance on STSS benchmark [20].

| | Conv. | Pub. Act. | Appo. | Inv. Com. | Online Act. | Help | AVG. |
|---|---|---|---|---|---|---|---|
| Base (GPT-4) | 48.6 | 59.6 | 1.2 | 2.3 | 63.4 | 61.5 | 39.4 |
| w. TDP [20] | 72.3 | 75.9 | 40.0 | 20.0 | 68.6 | 50.0 | 54.4 |
| w. HM [57] | 68.1 | 72.4 | 35.0 | 22.0 | 69.2 | 47.0 | 52.3 |
| w. ToM2C [58] | 70.2 | 74.1 | 38.0 | 18.0 | 66.5 | 52.0 | 53.1 |
| w. Generative Agents [28] | 65.4 | 70.3 | 42.0 | 19.0 | 67.8 | 55.0 | 53.3 |
| w. SymbolicToM [56] | 60.8 | 68.1 | 37.0 | 21.0 | 65.4 | 49.0 | 50.2 |
| **w. MetaMind (ours)** | **80.8** | **81.9** | **65.0** | **67.1** | **75.1** | **73.0** | **73.9** |

## 4.2 Social Cognition Task

We next evaluate MetaMind on a suite of social cognition tasks [19] designed to probe context-sensitive reasoning under social, cultural, and ethical norms. This benchmark includes eight real-world tasks such as Faux Pas Recognition (FRT), Scalar Implicature (SIT), and the Ambiguous Story Task (AST), which require models to interpret indirect social cues, detect norm violations, and reason about intent in nuanced interpersonal scenarios. These tasks are closely aligned with the functionality of the Moral Agent in Stage 2, which is responsible for refining mental-state hypotheses based on domain-specific constraints. We provide examples of these tasks in Appendix C.

As shown in Table 2, MetaMind yields consistent improvements across tasks, achieving **9%** improvement over the base model GPT-4 on average. Notably, we observe large gains in AST (+11.3%) and SIT (+11.4%), where MetaMind resolves contradictory cues (*e.g.*, sarcasm masked by polite wording) through hypothesis refinement, making vague social intentions clearer. A +10.6% gain in FRT suggests that the Moral Agent effectively prevents socially inappropriate interpretations by referencing implicit cultural rules. These results demonstrate that our multi-agent system excels in social cognition, especially when interpreting context-sensitive and norm-dependent cues. *Full results across LLM families are presented in Appendix B.1*.

## 4.3 Social Simulation Task (Open-Ended Generation)

To validate real-world applicability, we evaluate on Social Tasks in Sandbox Simulation (STSS) [20] - a benchmark testing goal-oriented social interaction across six domains: `Conversation`, `Public Activity`, `Appointment`, `Inviting Companions`, `Online Activity`, `Asking for Help`. As shown in Table 3, MetaMind achieves a remarkable **73.9%** average score, significantly outperforming GPT-4's 39.4%. MetaMind delivers substantial gains across all domains, including a +32.2% improvement in `Conversation`, where it maintains coherent character profiles across multi-turn interactions. A +22.3% boost in Public Activity, along with strong improvements in Appointment and Inviting, highlights its ability to track unstated user constraints—such as budget or schedule conflicts—through iterative, metacognitive reasoning.

We include results on **additional benchmarks**—such as SOTOPIA and SocialIQA—that further test open-ended interaction and commonsense social reasoning; detailed descriptions and evaluation setups are provided in Appendix A.6 and Appendix B.

Table 4: Ablation study on each component, evaluated on the social cognition tasks in [19].

**UOT**: Unexpected Outcome Test  **SIT**: Scalar Implicature Task  **PST**: Persuasion Story Task  **FBT**: False Belief Task
**AST**: Ambiguous Story Task  **HT**: Hinting Test  **SST**: Strange Story Task  **FRT**: Faux-pas Recognition Test

|  | UOT | SIT | PST | FBT | AST | HT | SST | FRT | Avg. |
|---|---|---|---|---|---|---|---|---|---|
| **MetaMind** | **81.5** | **60.4** | **64.8** | **90.1** | **88.8** | **86.2** | **88.4** | **83.9** | **80.5** |
| wo Stage 1 | 77.2 | 58.5 | 61.0 | 88.9 | 86.1 | 84.9 | 87.0 | 80.1 | 77.9 |
| wo Stage 2 | 75.6 | 57.8 | 59.3 | 88.1 | 84.7 | 84.0 | 86.2 | 78.4 | 76.7 |
| wo Stage 3 | 79.1 | 59.3 | 62.7 | 89.5 | 87.4 | 85.5 | 87.8 | 82.0 | 79.1 |
| wo SocialMemory | 73.9 | 56.2 | 58.1 | 87.4 | 82.3 | 83.1 | 85.0 | 76.8 | 75.4 |

Table 5: Ablation study on each component, evaluated on the social simulation task STSS [20].

|  | Conv. | Pub. Act. | Appo. | Inv. Com. | Online Act. | Help | Avg. |
|---|---|---|---|---|---|---|---|
| **MetaMind** | **80.8** | **81.9** | **65.0** | **67.1** | **75.1** | **73.0** | **73.9** |
| w.o Stage 1 | 78.1 | 78.4 | 59.0 | 60.3 | 72.1 | 62.3 | 68.3 |
| w.o Stage 2 | 79.2 | 79.3 | 61.7 | 62.2 | 73.7 | 67.0 | 70.5 |
| w.o Stage 3 | 58.7 | 67.2 | 54.2 | 43.2 | 61.9 | 61.7 | 57.8 |
| w.o SocialMemory | 70.5 | 72.3 | 57.0 | 58.0 | 64.8 | 61.2 | 63.9 |

# 5  Discussion

## 5.1  Ablation Study: Every Stage Matters

To validate the design of our staged multi-agent architecture, we conduct an ablation study isolating each core component of MetaMind. We evaluate performance drops when individual stages or mechanisms are removed.

**Stage 1 (Mental-State Reasoning)**: An important component in stage 1 is mental-state reasoning that generates structured hypotheses about the user's latent intent, emotion, or belief. As shown in Table 4, removing such structured reasoning leads to a **2.6%** drop on average across social cognition tasks, with substantial declines in high-ambiguity task like UOT (–4.3%). Similarly, in the STSS benchmark (Table 5), the performance is also impacted when we disable the mental-state reasoning. This highlights the central role of Stage 1 in enabling agents to hypothesize unspoken intentions.

**Stage 2 (Norm-Aware Refinement)**: Next, we ablate Stage 2, which is responsible for refining mental-state hypotheses using domain-specific rules such as cultural norms and ethical guidelines. Removing this component (wo Stage 2) is equivalent to directly passing the hypothesis from the Theory-of-Mind Agent to the Response Agent, without any constraint-driven refinement. As shown in Table 4, this results in a substantial **3.8%** drop in average performance on social cognition tasks. The degradation is most severe in tasks that involve norm violations or pragmatic interpretation, such as Faux-pas Recognition (–5.5%), where unrefined interpretations often lead to socially inappropriate or implausible responses.

**Stage 3 (Response via Validation)**: Finally, we ablate Stage 3, focusing on the impact of removing the validation mechanism within the Response Agent. This stage is responsible not only for generating a final response but also for validating it—ensuring consistency with the selected hypothesis, alignment with social memory, and appropriateness to the ongoing context. Skipping this step (Eq. 2) is equivalent to directly generating a response without any reflective checking. The effect is highly pronounced in the STSS benchmark (Table 5), where bypassing response validation leads to a **16.1%** drop in overall performance. Core categories such as Conversation (–22.1%), Behavior Appropriateness (–15.7%), and Help (–11.3%) suffer the most—highlighting how critical this final validation step is for delivering high-quality responses. These results demonstrate that MetaMind's staged architecture addresses distinct aspects of social intelligence: Stage 1 establishes core ToM competence, Stage 2 adapts reasoning to situational norms, and Stage 3 operationalizes this understanding in goal-oriented interactions. *No single component should be left out*, confirming that social intelligence requires layered cognitive architectures.

## 5.2  Comparison with Human Performance

How close do we stand with respect to human-level social reasoning performance? Figure 3 illustrates how MetaMind narrows the gap between LLMs and human-level Theory-of-Mind capabilities. In the left panel, we observe that baseline LLMs (without MetaMind) underperform across all six ToM

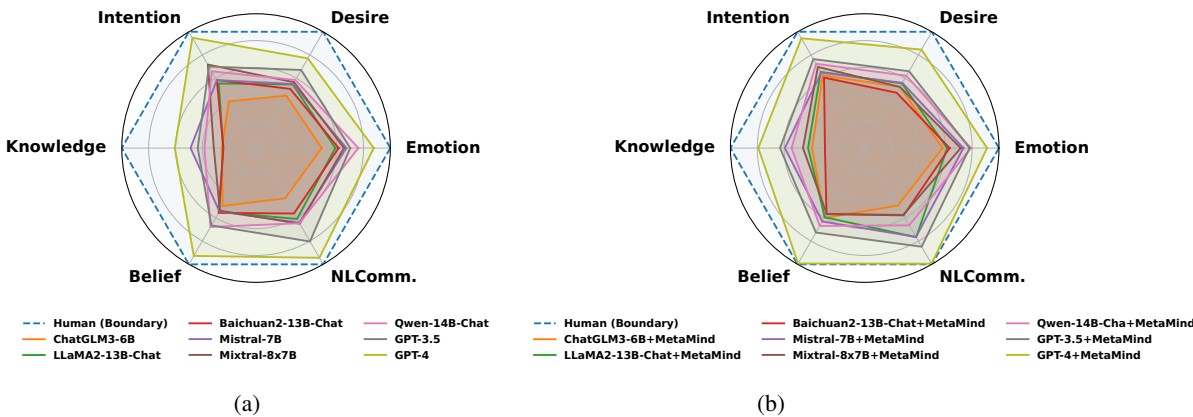

Figure 3: (a) Comparison of original LLMs and human capabilities. (b) Comparison between MetaMind-enhanced LLM performance against human capabilities.

dimensions. These models exhibit narrow capability profiles and struggle to generalize across the full space of human mental-state inference. In contrast, the right panel shows that after integrating MetaMind, LLMs expand their coverage significantly—demonstrating more human-like balance across all categories. *Notably, several models enhanced with MetaMind (e.g., GPT-4) approach human-level performance* in dimensions like Belief (89.3 vs 88.6), NL Communication (89.0 vs 88.5), and Desire (78.2 vs 76.5), indicating substantial improvement in nuanced social inference. These results confirm that MetaMind's structured, metacognitive reasoning framework enables LLMs to generalize beyond task-specific heuristics and approximate human social cognition more holistically.

## 5.3 Extension to Advanced Reasoning Models

MetaMind is designed to be model-agnostic and can be integrated with the latest state-of-the-art reasoning-capable LLMs. As shown in Table 6, MetaMind achieves strong gains when applied to frontier models such as DeepSeek-R1 [4], OpenAI o3 [59], and Claude 3.5 Sonnet [60], improving their ToM accuracy (average across 6 categories) even beyond existing high baselines. Notably, MetaMind boosts DeepSeek-R1 from 86.0% to 88.6%, and OpenAI o3 from 90.3% to 92.2%, demonstrating that even top-tier models benefit from metacognitive structuring.

Table 6: MetaMind boosts ToM performance of top-tier reasoning models.

| Model | Base | +MetaMind |
|---|---|---|
| Claude 3.5 Sonnet | 70.7 | **81.0** |
| DeepSeek-R1 | 86.0 | **88.6** |
| OpenAI o1 | 88.6 | **90.3** |
| OpenAI o3 | 90.3 | **92.2** |

These results confirm the compatibility and scalability of MetaMind across both proprietary and open-source systems.

## 5.4 Qualitative Case Studies

To further understand the strengths and weaknesses of our framework, we qualitatively analyze the intermediate outputs of MetaMind in six different scenarios, including *accommodation*, *collaboration*, *competition*, *exchange*, *negotiation*, and *persuasion*. We classify the cases into success and failure and provide in-depth analysis for each case. Due to space constraints, we provide details in Appendix C. We also conduct a human study and report our results in Appendix B.5. These case studies underscore MetaMind's capacity to simulate metacognitive reflection in real-world social contexts.

## 6 Conclusion

Human social intelligence hinges on the nuanced ability to infer unspoken mental states—a capability rooted in ToM that remains a critical gap in modern LLMs. To address this, we introduced MetaMind, a multi-agent framework inspired by metacognitive theories, which decomposes social reasoning into three collaborative stages: hypothesis generation, norm-aware refinement, and validated response generation. MetaMind enables adaptive and context-sensitive interactions that mirror human

metacognitive processes. Our experiments demonstrate that MetaMind achieves state-of-the-art performance across multiple social benchmarks. Notably, MetaMind enables LLMs to match human performance on key ToM tasks for the first time, bridging the gap between artificial and human social cognition. Ablation studies confirm the necessity of all components, underscoring the importance of structured hypothesis generation, ethical constraint enforcement, and iterative validation. We hope our framework advances applications in empathetic dialogue and culturally sensitive AI.

**Limitations.** While MetaMind achieves substantial gains, several challenges remain. First, Meta-Mind's performance depends on the quality of domain knowledge and the coverage of user context in memory; although effective in our experiments, broader deployment may require adaptation to diverse cultural norms and evolving social expectations. Moreover, MetaMind's performance is contingent on the backbone LLM's capabilities. While it improves various models, absolute performance gaps remain between small and large models. Lastly, existing benchmarks—though carefully curated—focus on constrained textual scenarios. Real-world social interactions involve multi-modal cues (tone, facial expressions), complex group dynamics, and long-term relationship building, which remain open challenges. Future work will explore expanding synthetic simulation environments and integrating more comprehensive ethical and cultural reasoning frameworks.

## Acknowledgement

We thank Sean Du, Froilan Choi, and Pengyue Jia for their valuable suggestions on the draft and Yonghang Chen for his contribution to the website.

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

**Appendix**

# Contents

# A   Implementation Details

## A.1   Stage 1: Generating Mental State Hypothesis via Theory-of-Mind Agent

---

**Contextual Analysis Task**

**Input**:
- Analyze the user's current statement: `[u_t]`
- Within conversational context: `[C_t]`

**Objective**: Generate 3–5 commonsense interpretations of the user's unstated needs by:

1. Identifying key semantic triggers in the utterance
2. Mapping these triggers to plausible psychosocial motivations
3. Considering cultural and linguistic norms for indirect communication

**Output Format**:
- Interpretation 1: `[Explanation]` (*Contextual Support*: `[Relevant C_t Excerpt]`)
- Interpretation 2: `[...]`, etc.

**Memory Integration Task**

**Input**:
- Proposed hypothesis: `[Selected Common-Sense Interpretation]`
- Social memory database: `[M_t]`

**Step 1: Identify memory matching criteria**

---

- *Emotional patterns*: [List relevant M_t emotion tags]
- *Behavioral history*: [Past interactions demonstrating similar intent]
- *Preference alignment*: [Stated/implied user preferences]

**Step 2: Calculate hypothesis validity score (1-5)**

- Consistency with historical patterns
- Absence of contradictory evidence
- Temporal relevance

**Output Format**: *"Hypothesis [X] shows [strong/weak] memory alignment (Score: [N]). Key corroborations: [List]."*

## Mental State Typology Task

**Processed Input**:

- Utterance: [u_t]
- Top Hypothesis: [Interpretation]
- Memory Correlations: [Findings]

**Classification Markers**:

- **Belief**: Cognitive representations of reality
- **Desire**: Preferences or goal states
- **Intention**: Action-oriented plans
- **Emotion**: Affective states
- **Thought**: Conscious reasoning processes

**Output Format**:

- Primary Marker: [T] (Confidence: [%])
  *Rationale*: Psychological justification using Fiske's social cognition framework
- Secondary Markers: [List]
  *Interaction Effects*: How the markers co-influence the hypothesis

## Mental State Space Planning

**Parameters**:

- Target diversity: 40% across marker types
- Hypothesis count: $k = $ [N]
- Evidence threshold: Medium–High confidence

**Guidelines for Generation**:

1. For each identified marker type [T], generate 1–2 extra hypotheses
2. Ensure orthogonal reasoning paths across hypotheses
3. Include both surface-level interpretations and deep psychosocial explanations

**Output Format for Each Hypothesis**:

- [Hypothesis #]:
  **Type**: [Belief/Desire/Intention/Emotion/Thought]
  **Description**: Two-sentence natural language explanation
  **Evidential Basis**:
    - Linguistic Signals: [Lexical/paralinguistic features]
    - Contextual Drivers: [C_t elements]
    - Memory Anchors: [M_t correlations]

## A.2 Stage 2: Refining Hypothesis via Moral Agent

The Moral Agent refines the latent mental state hypotheses produced by the ToM Agent by enforcing domain–specific constraints and selecting the most contextually plausible yet information-rich

interpretation. The hypothesis selection system is summarized in Alg. 1; detailed prompt templates for constraint refinement are shown below.

---

**Moral Constraint Refinement Task**

**Input**

- *Hypothesis:* [h_i]
- *Domain Rule Type:* [Cultural / Ethical / Role-Based]
- *Constraint Specifications:* [Relevant $\mathcal{D}$ rules]

**Step 1: Constraint Identification**

- Flag elements violating [Domain Rule Type] norms.
- Highlight ambiguous social signals requiring disambiguation.

**Step 2: Re-interpretation Protocol**

- *Cultural*: Remap interpretations via Hofstede's cultural–dimension framework.
- *Ethical*: Apply IEEE *EthicallyAlignedDesign* principles.
- *Role-Based*: Enforce Goffman's facework theory on role-appropriate behavior.

**Step 3: Tone Alignment**

- Appropriateness scaling (1=informal, 5=formal).
- Politeness markers from Brown&Levinson's theory.

**Output**

- *Original Hypothesis*: [h_i]
- *Revised Hypothesis* ($\tilde{h}_i$): [Socially compliant interpretation]
- *Modification Log*:
  - Constrained Elements: [List]
  - Applied Transformations: [Techniques]
  - Residual Risk Assessment: [Concerns]

---

**Hypothesis Selection.** Given candidate hypotheses $\{\tilde{h}_1, \ldots, \tilde{h}_k\}$ and weights $\lambda = 0.6$, The Moral Agent computes a composite score for each candidate as described in Alg. 1 and selects $\tilde{h}^* = \arg\max_i s_i$ for downstream response generation.

---

**Few-shot Prompt for $\mathcal{M}$ (logit-free)**

SYSTEM: You are an expert social–context evaluator. Given **Social Context**, **Social Memory**, **User Prompt** and a candidate **Hypothesis**, respond with ''high'', ''mid'', or ''low'' to indicate the likelihood that the hypothesis correctly interprets the user's latent mental state.

=== QUERY ===
**Social Context**: $\{C\}$
**Social Memory**: $\{M\}$
**user Prompt**: $\{u\}$
**Hypothesis**: $\{h\}$
**Rating**: ___________

---

### A.3 Stage 3: Generating and Validating Output via Response Agent

---

**Contextualized Response Synthesis**

**Input Parameters**

- **Selected Hypothesis:** [$\tilde{h}$] (Type: T)
- **Social Memory Profile:** [M_t]
- **User Prompt:** [u_t]

---

**Algorithm 1:** Hypothesis Selection

**Input:** Candidate hypotheses $\mathcal{H}_t = \{\tilde{h}_1, \ldots, \tilde{h}_k\}$;
User prompt $u_t$, social context $C_t$, social memory $M_t$;
Conditional LM $\mathcal{M}_{\text{context}}$ for $P(h \mid u_t, C_t, M_t)$;
Prior LM $\mathcal{M}_{\text{prior}}$ for $P(h)$;
Weight $\lambda$.

**Output:** Selected hypothesis $\tilde{h}^*$

1 **foreach** $\tilde{h}_i \in \mathcal{H}_t$ **do**

2      $P_{\text{cond}} \leftarrow \texttt{ConditionalProb}(\mathcal{M}_{context}, \tilde{h}_i, u_t, C_t, M_t)$;

3      $P_{\text{prior}} \leftarrow \texttt{PriorProb}(\mathcal{M}_{prior}, \tilde{h}_i)$;
     ; // Information gain as log-likelihood shift

4      $IG_i \leftarrow \log(P_{\text{cond}} + \varepsilon) - \log(P_{\text{prior}} + \varepsilon)$;
     ; // Composite score

5      $s_i \leftarrow \lambda \cdot P_{\text{cond}} + (1 - \lambda) \cdot IG_i$;

6 $\tilde{h}^* \leftarrow \arg\max_{\tilde{h}_i \in \mathcal{H}_t} s_i$;

7 **return** $\tilde{h}^*$;

---

8 **Subroutines: Function** $\texttt{ConditionalProb}(\mathcal{M}, h, u, C, M)$:

9      **if** $\mathcal{M}$ *exposes token logits* **then**

10          Prompt $\leftarrow \textsc{Concat}(\langle\text{USR}\rangle u, \langle\text{CON}\rangle C, \langle\text{MEM}\rangle M)$;

11          Tokenize $h \rightarrow (h_1, \ldots, h_L)$;

12          $\ell \leftarrow 0$;

13          **for** $n \leftarrow 1$ **to** $L$ **do**

14              $\mathbf{z}_n \leftarrow \mathcal{M}(\text{Prompt}\|h_{<n})$;

15              $\log p_n \leftarrow z_{n,h_n} - \log \sum_j \exp(z_{n,j})$;

16              $\ell \leftarrow \ell + \log p_n$;

17          **return** $\exp(\ell)$ ;                      // $P(h \mid \text{Prompt})$

18      **else**

19          rating $\leftarrow \mathcal{M}(\text{few-shot prompt})$;

20          **return** $\textsc{Map}(\text{rating})$ ;             // e.g., *high* $\mapsto$ 0.9

21 **Function** $\texttt{PriorProb}(\mathcal{M}, h)$:

22      **return** $\texttt{ConditionalProb}(\mathcal{M}, h, \emptyset, \emptyset, \emptyset)$

---

**Generation Protocol**

1. **Tone Calibration** Map emotional tone using Plutchik's emotion wheel, leveraging:
   - Emotion tags from the selected hypothesis
   - Historical emotional patterns in `M_t`

2. **Memory Integration** Incorporate up to three memory anchors:
   - **Preference:** `"User's stated preference"`
   - **Behavioral Pattern:** `"Recurring interaction motif"`
   - **Emotional Baseline:** `"Characteristic emotional state"`

3. **Pragmatic Realization** Construct the response by applying:
   - **Speech Act Design:** Searle's taxonomy (Assertive, Directive, Commissive)
   - **Politeness Strategy:** Brown & Levinson's face-management theory
   - **Cohesion Devices:** Halliday's systemic functional linguistics

**Output Requirements**

- **Primary response (`o_t`):** Natural language implementation
- **Generation Metadata:**
  - **Emotional Valence:** Arousal–Dominance–Valence (ADV) scores

> – **Memory Utilization:** Specific `M_t` elements used
> – **Hypothesis Fidelity:** Percentage match to selected hypothesis

**Response Quality Audit: Validation Rubric**

**Input Parameters**

- **Response:** [o_t]
- **Selected Hypothesis:** [$\tilde{h}$] (Type: T)
- **Social Memory Profile:** [M_t]
- **Social Context:** [C_t]
- **User Utterance:** [u_t]
- **Tradeoff Weight:** [$\beta$]

**A. Empathy Assessment**

1. **Affective Alignment (40%):**
   - Match emotional trajectory to response emotion markers
   - Analyze lexical affect to current status via NRC Emotion Lexicon

2. **Cognitive Resonance (60%):**
   - Presence of perspective-taking markers (*e.g.*, "I understand...")
   - Accommodation of `M_t`-based user preferences

**B. Coherence Evaluation**

1. **Contextual Continuity (50%):**
   - Referential consistency using Centering Theory
   - Temporal/causal coherence with prior dialogue

2. **Hypothesis Congruence (50%):**
   - Propositional alignment via Semantic Role Labeling (SRL)
   - Hypothesis-driven content anchoring with cross-modal consistency

**Scoring Protocol**

- Sub-score each category on a 0-1 scale
- Compute:

$$\text{Empathy} = 0.4 \cdot \text{A1} + 0.6 \cdot \text{A2} \qquad \text{Coherence} = 0.5 \cdot \text{B1} + 0.5 \cdot \text{B2}$$

$$\text{Final Utility: } \mathcal{U} = \beta \cdot \text{Empathy} + (1 - \beta) \cdot \text{Coherence}$$

**Output Format:**

> `Validation Report` **Empathy Score:** [X/1] (Strengths: [...])
> **Coherence Score:** [Y/1] (Weaknesses: [...])
> **Total Utility:** [$\mathcal{U}$] → [Acceptable / Marginal / Unacceptable]

**Response Optimization Protocol**

**Trigger:** If $\mathcal{U} < 0.9$
**Create priors and trace back to the Planning Stage**

1. **Empathy Boosting (Decety's Model):**
   - Add reverse affective perspective-taking markers
   - Refine emotional expression with Barrett's conceptual act theory

2. **Coherence Restoration (Grosz's Model):**
   - Insert retrospective cues (*e.g.*, "As we discussed...")
   - Add prospective markers (*e.g.*, "Moving forward...")

3. **Memory Reinforcement:**
   - Insert additional `C_t`/`M_t` references using:

> – Episodic framing: "Last time you mentioned..."
> – Preference justification: "Knowing you prefer..."
>
> **Iteration Limit:** 3 revisions **Termination Condition:** $\mathcal{U} \geq 0.9$ or maximum reached
> **Output Documentation**
> - **Revision History:** [List of edits per iteration]
> - **Final Utility Score:** $\mathcal{U}$
> - **Residual Risk Statement:** [Unresolved issues or caveats]

## A.4 Social Memory

Social Memory ($M_t$) is a dynamic, structured knowledge base that evolves across interactions to capture long-term user patterns, social norms, and feedback-based adjustments. It is designed around three core principles: (1) Grounding in context: Memory is initialized based on the situation and roles involved in the interaction. (2) Updating through user modeling: Updating memory via validated interpretations of user mental states. (3) Improving through feedback: It incorporates signals from failures or corrections to better guide future responses.

**Initialization: Context-Aware Memory ($M_0$)**   The initial memory $M_0$ is constructed using the interaction scenario, including setting (*e.g.*, professional workspace vs. casual social setting); role relationships (*e.g.*, doctor-patient hierarchy, friend-friend reciprocity), and cultural/ethical expectations.

**Memory Update ($M_t \rightarrow M_{t+1}$)**   At each turn $t$, the memory is updated using the long-term hypothesis $h'$ generated by the ToM Agent, which summarizes persistent user states as:

$$h' = \{\text{Beliefs}(u_t), \text{Desires}(u_t)\} \cup \{\text{Emotions}(u_t) \cap \text{Emotion Patterns}(u_{1:t})\},$$

where Emotion Patterns represent recurring affective tendencies (*e.g.*, a user who frequently expresses frustration during task-related conversations).

**Feedback-Based Correction.**   If the system's response $o_t$ is unsuccessful, such as receiving low utility feedback or being flagged by a user or evaluator, memory is adjusted. The evaluator's critique (*e.g.*, "overly formal tone") is mapped to a structured emotional pattern $E_i$. If this contradicts existing memory, the system lowers the weight of the old pattern. Otherwise, $E_i \rightarrow M_{t+1}$.

## A.5 Hyperparameters and Configurations

**Experimental Setup.**   We conduct a comprehensive grid search to optimize MetaMind's key parameters. Specifically, we sweep over the hypothesis size $k \in \{0, 1, \ldots, 10\}$, the coefficient $\lambda \in [0, 1]$ (in steps of 0.01), and the balance factor $\beta \in [0, 1]$ (in steps of 0.01).[2] We use GPT-4 as the underlying model and report overall accuracy on TOMBENCH as the evaluation metric.

**Sensitivity Analysis.**   The *global* optimum is found at $(k^\star, \lambda^\star, \beta^\star) = (6, 0.64, 0.78)$, achieving an overall accuracy of **0.822**. Notably, the most consistently high-performing window sizes are $k \in \{6, 7, 8\}$. Figure 4a to 4c visualize the smoothed accuracy landscapes over the $(\lambda, \beta)$ grid for each of these values of $k$.

**Final Configuration.**   To reduce inference overhead, we select the *smaller* window size $k=6$ for all experiments while fixing $(\lambda, \beta) = (0.60, 0.80)$, which lies on the high-accuracy ridge close to the global optimum.[3]

## A.6 More Details of the Benchmarks

We evaluate MetaMind on four benchmarks: ToMBench, STSS, SocialIQA, and SOTOPIA. These four datasets collectively span multiple-choice reasoning, open-ended interaction, and grounded action execution, giving a broad measure of MetaMind's social intelligence beyond standard QA.

---

[2]This results in a total of $11 \times 101 \times 101 = 112{,}211$ configurations.
[3]All numbers are tested on a single A100 80GB for 166.8 hours; batch size = 1.

Table 7: Best configuration for each $k$ on TOMBENCH.

| $k$ | $\lambda^\star$ | $\beta^\star$ | Overall Accuracy |
|---|---|---|---|
| 0 | 0.00 | 0.36 | 0.420 |
| 1 | 0.06 | 0.35 | 0.451 |
| 2 | 0.62 | 0.72 | 0.503 |
| 3 | 0.62 | 0.77 | 0.587 |
| 4 | 0.66 | 0.78 | 0.672 |
| 5 | 0.64 | 0.79 | 0.756 |
| 6 | 0.64 | 0.78 | **0.822** |
| 7 | 0.64 | 0.80 | 0.755 |
| 8 | 0.63 | 0.76 | 0.672 |
| 9 | 0.63 | 0.74 | 0.587 |
| 10 | 0.65 | 0.74 | 0.503 |

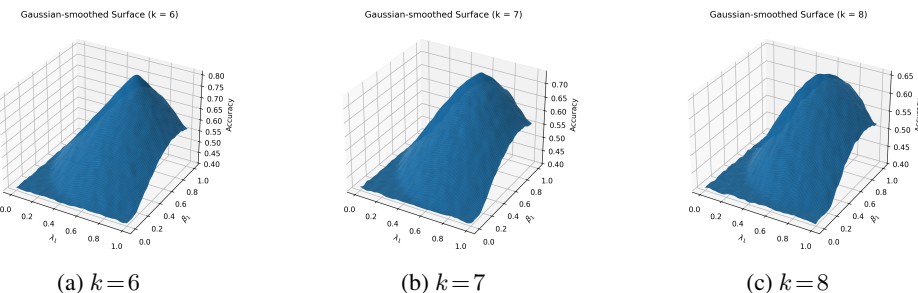

(a) $k=6$        (b) $k=7$        (c) $k=8$

Figure 4: Accuracy landscape over $(\lambda, \beta)$ for the three most competitive $k$.

**ToMBench**[4] offers the most comprehensive multiple-choice evaluation of Theory-of-Mind, covering 8 distinct ToM reasoning tasks (*e.g.*, first-/second-order false belief, emotion attribution) and 31 fine-grained social-cognitive abilities. All 1,079 test items are author-written to avoid training leakage and are accompanied by gold-standard answers, allowing for fully automated accuracy evaluation. Following the original protocol, we evaluate performance on the full test set.

**STSS**[5] is an action-level benchmark that evaluates whether language agents can successfully achieve social goals in a multi-agent sandbox environment. The suite instantiates 30 task templates spanning 5 categories: `Public Activity`, `Appointment`, `Inviting Companions`, `Online Activity`, and `Asking for Help`. Tasks are instantiated within the Smallville simulator and evaluated using objective metrics such as guest count or goal completion rate. We evaluate MetaMind on the full test set, including the conversation-focused split, comprising 30 episodes (5 per category), and report the normalized success score.

**SocialIQA**[6] probes models' ability to infer motivations, reactions, and mental states in everyday situations. The dataset consists of over 38,000 multiple-choice questions, each comprising a context, a question, and three answer choices. Following standard protocol, we evaluate MetaMind on the full test set and report multiple-choice accuracy, using leaderboard-reported LLM performance as the baseline for comparison.

**SOTOPIA**[7] is an open-ended role-play environment containing 90 social scenarios and 40 richly annotated characters; each episode asks two language agents to pursue private yet potentially conflicting social goals through language, gestures, and actions. Performance is evaluated across seven social dimensions: *Believability* (BEL, naturalness and persona consistency), *Relationship* (REL, whether rapport improves), *Knowledge Acquisition* (KNO, curiosity and information gain), *Secret Keeping* (SEC, leakage penalties,), *Social-Rule Compliance* (SOC, norm/legal violations), *Financial/-Material Benefits* (FIN, economic payoff), and *Goal Completion* (GOAL, task success). Following

[4] https://github.com/zhchen18/ToMBench
[5] https://github.com/wcx21/Social-Tasks-in-Sandbox-Simulation
[6] https://huggingface.co/datasets/allenai/social_i_qa
[7] https://huggingface.co/datasets/cmu-lti/sotopia

the official evaluation setup, we use GPT-4 as the automatic judge—validated against human ratings for most dimensions—and compute the *Overall* score as the range-normalized average across all seven dimensions.

## A.7  More Details of the Baselines

We compare MetaMind with five baselines: Generative Agents, CoT prompting, SymbolicToM, HM, and ToM2C. These five baselines collectively span prompt-engineering, symbolic reasoning, hybrid LLM+RL, and fully agentic memory systems.

**Generative Agents**[8] extend an LLM with three modules—observation, reflection, and planning—plus a long-term, natural-language memory, producing sandbox characters that wake up, pursue daily goals, and initiate free-form social interactions in a "Smallville" town simulation. The official code base exposes a REST API that receives a textual situation and returns the agent's next action and optional memory updates. We keep the authors' memory–retrieval stack (300-slot episodic buffer, cosine-similarity retriever), and call a maximum of three "reflection cycles" after each story paragraph. All other hyperparameters follow the 'town_v1' config in the repo.

**CoT prompting** [31] appends a few-shot chain-of-thought demonstration—"Let's think step by step. . . "—to the user query, inducing the LLM to emit intermediate reasoning before the final answer. We adopt eight exemplars drawn from the authors' public GSM8K prompt and adapt them to social-reasoning form. We sample five reasoning paths with temperature 0.7 and use self-consistency to select the majority answer, matching the hyper-parameter setting in the original paper for tasks with open-ended answers.

**SymbolicToM**[9] is a decoding-time wrapper that constructs a multi-layer belief graph and feeds the current belief state along with the question back into the base LLM, greatly boosting zero-shot ToM performance on ToMi and related benchmarks. We port the author-released JAX implementation to Python 3.11 and limit graph depth to second-order beliefs. We follow the authors' decoding scheme with top-$p$ 0.95 and max-tokens 128.

**HM**[10] equips an LLM agent with a Hypothesis Generator and a Hypothesis Critic that iteratively propose and test natural-language theories of the other agents' strategies during Melting-Pot games, yielding large gains over both MARL and script-based agent baselines. We reuse the official checkpoint trained on the Competitive Stag Hunt scenario. For single-shot questions, we run one hypothesis-generation round with 3 candidates and pick the answer derived from the highest-scoring hypothesis according to the Critic. Temperature 0.3 and KL-penalty 0.2 mirror the authors' ablation-best setting.

**ToM2C**[11] introduces hierarchical agents that infer others' latent goals, decide when/whom to communicate, and then plan sub-goals for cooperative navigation and multi-sensor target coverage. A theory-of-mind module parameterizes a Bayesian latent-goal predictor that is updated online. For text-based benchmarks, we follow recent practice and verbalize the latent-goal vector through a template sentence ("I believe the user wants . . . "), feeding that description plus the original narrative into GPT-4 to obtain a final answer.

# B  Additional Results

## B.1  Theory-of-Mind Reasoning

In Table 8, we report full results across 16 diverse LLM backbones on the ToMBench benchmark, measuring six fine-grained Theory-of-Mind abilities. MetaMind consistently enhances ToM reasoning across all models, with average accuracy gains of +10.2% in low-resource models like ChatGLM-6B, halving the gap to GPT-3.5. Notably, high-capacity models like DeepSeek R1, Grok-3, OpenAI o1/o3, GPT-4 now surpass human performance on $\geq$ 3 abilities (asterisked in Table 8), most prominently NL Communication, Belief, and Desire. We additionally report performance using vanilla Chain-of-

---

[8]https://github.com/joonspk-research/generative_agents
[9]https://github.com/msclar/symbolictom
[10]https://github.com/locross93/Hypothetical-Minds
[11]https://github.com/UnrealTracking/ToM2C

Table 8: Ability-oriented ToM performance in accuracy. "LLM Grand Mean" is the average performance of all 16 LLMs. * represents that this dimension has exceeded human behavior.

| SUBJECT | Emotion | Desire | Intention | Knowledge | Belief | NL Comm. | AVG. |
|---|---|---|---|---|---|---|---|
| **Human** | **86.4** | **78.2** | **90.4** | **82.2** | **89.3** | **89.0** | **86.1** |
| ChatGLM3-6B | 42.2 | 40.7 | 35.9 | 22.0 | 44.5 | 38.5 | 37.3 |
| ChatGLM3-6B + CoT | 46.7 | 43.7 | 49.8 | 28.9 | 48.6 | 40.1 | 43.0 |
| **ChatGLM3-6B + MetaMind** | **50.1** | **47.5** | **55.8** | **33.7** | **53.6** | **44.0** | **47.5** |
| LLaMA2-13B-Chat | 51.0 | 49.4 | 49.6 | 21.1 | 49.0 | 54.3 | 45.7 |
| LLaMA2-13B-Chat + CoT | 48.1 | 44.9 | 51.7 | 30.7 | 47.9 | 62.7 | 47.7 |
| **LLaMA2-13B-Chat + MetaMind** | **53.0** | **50.2** | **58.5** | **36.3** | **53.1** | **68.2** | **53.4** |
| Baichuan2-13B-Chat | 53.1 | **46.0** | 52.2 | 20.9 | 49.8 | 50.1 | 45.4 |
| Baichuan2-13B-Chat + CoT | 49.7 | 37.5 | 47.8 | 19.3 | 45.2 | 47.5 | 41.2 |
| **Baichuan2-13B + MetaMind** | **54.6** | 43.0 | **54.1** | **25.5** | **50.8** | **51.5** | **47.0** |
| Mistral-7B | 58.1 | 49.8 | 52.2 | 42.0 | 48.7 | 57.2 | 51.3 |
| Mistral-7B + CoT | 57.9 | 45.1 | 51.1 | 44.5 | 50.1 | 62.4 | 51.9 |
| **Mistral-7B + MetaMind** | **63.5** | **50.7** | **57.9** | **51.2** | **56.4** | **68.0** | **58.2** |
| Mistral-8x7B | 56.6 | **51.2** | **64.1** | 27.1 | 48.1 | **57.9** | 50.8 |
| Mistral-8x7B + CoT | 56.0 | 41.5 | 55.3 | 33.2 | 44.3 | 45.5 | 46.0 |
| **Mistral-8x7B + MetaMind** | **61.8** | 47.5 | 62.3 | **39.6** | **50.5** | 51.2 | **52.0** |
| Qwen-14B-Chat | 65.8 | 52.9 | 58.9 | 33.1 | **60.6** | 57.5 | 54.8 |
| Qwen-14B-Chat + CoT | 62.7 | 50.2 | 57.8 | 40.1 | 53.6 | 53.2 | 52.9 |
| **Qwen-14B-Chat + MetaMind** | **68.3** | **56.2** | **64.7** | **46.9** | 59.8 | **59.1** | **59.4** |
| GPT-3.5-Turbo-0613 | 65.6 | 53.4 | 61.0 | 36.3 | 61.4 | 66.9 | 57.4 |
| GPT-3.5-Turbo-0613 + CoT | 62.7 | 52.1 | 63.8 | 43.3 | 58.7 | 71.6 | 58.7 |
| **GPT-3.5-Turbo-0613 + MetaMind** | **68.1** | **57.8** | **68.5** | **48.5** | **63.9** | **76.2** | **64.2** |
| GPT-3.5-Turbo-1106 | 60.6 | **60.7** | 62.6 | 37.4 | 59.4 | 71.5 | 58.7 |
| GPT-3.5-Turbo-1106 + CoT | 62.3 | 54.7 | 63.1 | 49.6 | 59.9 | 70.8 | 60.1 |
| **GPT-3.5-Turbo-1106 + MetaMind** | **67.5** | 59.7 | **68.3** | **54.3** | **64.9** | **75.5** | **65.1** |
| Claude-3.5 Sonnet | 68.0 | 58.2 | 78.0 | 43.0 | 76.0 | 77.0 | 66.7 |
| Claude-3.5 Sonnet + CoT | 66.0 | 55.0 | 76.0 | 40.0 | 75.0 | 76.0 | 64.7 |
| **Claude-3.5 Sonnet + MetaMind** | **72.5** | **63.5** | **80.5** | **47.5** | **79.5** | **80.5** | **70.7** |
| DeepSeek v3 | 70.0 | 60.0 | 80.0 | 45.0 | 78.0 | 79.0 | 68.7 |
| DeepSeek v3 + CoT | 68.0 | 57.0 | 78.0 | 42.0 | 77.0 | 78.0 | 66.0 |
| **DeepSeek v3 + MetaMind** | **74.5** | **65.5** | **82.5** | **49.5** | **81.5** | **82.5** | **72.7** |
| GPT-4-0613 | 72.0 | 60.2 | 66.1 | 48.1 | 76.1 | 81.5 | 67.3 |
| GPT-4-0613 + CoT | 73.1 | 67.1 | 57.5 | 57.5 | 76.4 | 82.2 | 71.3 |
| **GPT-4-0613 + MetaMind** | **77.9** | **72.2** | **76.9** | **63.1** | **81.6** | **86.5** | **76.2** |
| GPT-4-1106 | 75.7 | 69.7 | **84.7** | 52.1 | 82.8 | 84.0 | 74.8 |
| GPT-4-1106 + CoT | 73.2 | 63.3 | 77.9 | 60.4 | 83.6 | 83.0 | 73.6 |
| **GPT-4-1106 + MetaMind** | **78.7** | **76.5** | 84.3 | **68.2** | **88.6** | **88.5** | **81.0** |
| DeepSeek R1 | 84.0 | 77.0 | 88.5 | 60.0 | 89.0 | 90.0 | 81.4 |
| DeepSeek R1 + CoT | 82.0 | 74.5 | 86.0 | 57.5 | 88.0 | 88.5 | 79.4 |
| **DeepSeek R1 + MetaMind** | **86.0** | **79.3*** | **90.2** | **63.2** | **91.1*** | **92.0*** | **83.6** |
| Grok-3 Think | 86.0 | 79.0 | 91.0 | 65.0 | 91.0 | 91.5 | 83.9 |
| Grok-3 Think+ CoT | 84.0 | 76.5 | 88.5 | 62.5 | 90.0 | 90.0 | 81.9 |
| **Grok-3 Think+ MetaMind** | **88.3*** | **81.3*** | **92.3*** | **68.2** | **92.9*** | **93.2*** | **86.0** |
| OpenAI o1 | 88.2 | 82.5 | 93.1 | 68.5 | 93.0 | 93.5 | 86.5 |
| OpenAI o1 + CoT | 86.0 | 80.0 | 90.0 | 66.0 | 92.0 | 92.5 | 84.4 |
| **OpenAI o1 + MetaMind** | **90.4*** | **85.3*** | **94.5*** | **72.0** | **94.6*** | **94.8*** | **88.6*** |
| OpenAI o3 | 90.4 | 85.1 | 95.3 | 80.8 | 95.0 | 95.2 | 90.3 |
| OpenAI o3 + CoT | 88.0 | 82.0 | 92.1 | 78.0 | 93.7 | 94.0 | 88.0 |
| **OpenAI o3 + MetaMind** | **92.3*** | **87.5*** | **96.5*** | **84.0*** | **96.4*** | **96.3*** | **92.2*** |
| LLM Grand Mean | 68.0 | 61.0 | 69.6 | 43.9 | 68.9 | 71.6 | 63.8 |
| LLM Grand Mean + CoT | 66.6 | 57.8 | 68.8 | 47.1 | 67.8 | 71.1 | 63.2 |
| **LLM Grand Mean + MetaMind** | **71.7** | **64.0** | **74.2** | **53.2** | **72.4** | **75.5** | **68.6** |

Thought (CoT) prompting, which slightly degrades the overall performance (–1.2%) due to unguided reasoning. MetaMind reverses this trend with structured reasoning, indicating that its metacognitive system is more reliable. Overall, our results validate MetaMind as a general-purpose strategy for social reasoning, with applicability across both weak and strong foundation models.

## B.2 Social Cognition

In Table 9, we report the full performance on eight challenging social cognition tasks such as False Belief, Scalar Implicature, and Persuasion from the ToMBench suite. MetaMind yields significant improvements across 16 tested LLMs, with average accuracy improved by 5.3%.

Table 9: Task-oriented ToM performance in accuracy. "LLM Grand Mean" is the average performance of all 16 LLMs. * represents that this dimension has exceeded human behavior.

**UOT**: Unexpected Outcome Test  **SIT**: Scalar Implicature Task  **PST**: Persuasion Story Task  **FBT**: False Belief Task
**AST**: Ambiguous Story Task  **HT**: Hinting Test  **SST**: Strange Story Task  **FRT**: Faux-pas Recognition Test

| SUBJECT | UOT | SIT | PST | FBT | AST | HT | SST | FRT | AVG. |
|---|---|---|---|---|---|---|---|---|---|
| **Human** | **89.3** | **75.5** | **70.0** | **86.8** | **95.0** | **97.1** | **89.2** | **80.4** | **85.4** |
| ChatGLM3-6B | 44.3 | 28.0 | 41.0 | 48.5 | 41.0 | 36.9 | 37.8 | 44.6 | 40.3 |
| ChatGLM3-6B + CoT | 50.3 | 26.5 | 41.0 | 51.2 | 44.0 | 42.7 | 44.2 | 51.4 | 43.9 |
| **ChatGLM3-6B + MetaMind** | **55.1** | **31.2** | **45.2** | **62.1** | **52.7** | **46.9** | **61.8** | **71.9** | **53.4** |
| LLaMA2-13B-Chat | 52.7 | 23.5 | **43.0** | 42.8 | 47.5 | **48.5** | 58.0 | 58.4 | 46.8 |
| LLaMA2-13B-Chat + CoT | 52.7 | 23.5 | 39.0 | 43.0 | 48.5 | 43.7 | 59.5 | 62.1 | 46.5 |
| **LLaMA2-13B-Chat + MetaMind** | **57.9** | **28.1** | 42.9 | **47.3** | **53.4** | 48.1 | **65.5** | **68.3** | **51.4** |
| Baichuan2-13B-Chat | 53.7 | **32.0** | 36.0 | 51.5 | 50.5 | 58.3 | 50.4 | 61.3 | 49.2 |
| Baichuan2-13B-Chat + CoT | 48.7 | 23.0 | 34.0 | 44.2 | 44.0 | 49.5 | 51.1 | 52.5 | 47.7 |
| **Baichuan2-13B + MetaMind** | **59.7** | 29.2 | **37.4** | 49.3 | 56.7 | **58.7** | 58.1 | **71.9** | **52.6** |
| Mistral-7B | 58.0 | **34.5** | **51.0** | 46.7 | 51.0 | **43.7** | 60.0 | 66.8 | 51.5 |
| Mistral-7B + CoT | 55.3 | 28.0 | 42.0 | 47.0 | 46.5 | 37.9 | 63.4 | 64.1 | 48.0 |
| **Mistral-7B + MetaMind** | **67.1** | 30.8 | 50.6 | **51.9** | **51.7** | 41.7 | **69.7** | **70.7** | **54.3** |
| Mistral-8x7B | 58.7 | 42.5 | **55.0** | 37.8 | 69.5 | 55.3 | **53.8** | 54.1 | 53.3 |
| Mistral-8x7B + CoT | 52.3 | 29.5 | 39.0 | 43.8 | 59.5 | 54.4 | 39.8 | 54.3 | 46.6 |
| **Mistral-8x7B + MetaMind** | **71.8** | **49.5** | 45.1 | **59.1** | **72.6** | **59.8** | 48.1 | **59.7** | **58.2** |
| Qwen-14B-Chat | 63.7 | 30.5 | **51.0** | 58.7 | 64.0 | **56.3** | 59.5 | 69.5 | 56.7 |
| Qwen-14B-Chat + CoT | 58.0 | 31.0 | 44.0 | 54.7 | 63.0 | 48.5 | 53.6 | 67.7 | 52.6 |
| **Qwen-14B-Chat + MetaMind** | **71.8** | **34.7** | 49.5 | **60.2** | **69.3** | 53.4 | **66.2** | **77.8** | **60.4** |
| GPT-3.5-Turbo-0613 | 63.3 | **35.0** | 49.0 | 62.3 | 63.5 | **53.4** | 66.1 | 67.0 | 57.5 |
| GPT-3.5-Turbo-0613 + CoT | 58.3 | 26.5 | 48.0 | 64.0 | 63.0 | 41.7 | 66.8 | 70.4 | 54.2 |
| **GPT-3.5-Turbo-0613 + MetaMind** | **68.5** | 33.0 | **52.8** | **70.4** | **64.3** | 45.9 | **78.4** | **77.6** | **61.4** |
| GPT-3.5-Turbo-1106 | 66.0 | 33.0 | 56.0 | 55.0 | 60.5 | **64.1** | 69.0 | 72.5 | 59.5 |
| GPT-3.5-Turbo-1106 + CoT | 64.7 | 35.0 | 54.0 | 56.3 | 63.0 | 51.5 | 68.6 | 70.9 | 58.0 |
| **GPT-3.5-Turbo-1106 + MetaMind** | **75.6** | **38.5** | **59.4** | **63.3** | **69.3** | 56.7 | **78.4** | **79.9** | **65.1** |
| Claude-3.5 Sonnet | 67.1 | 45.8 | **60.4** | 85.1 | 73.3 | 78.6 | 80.3 | 69.5 | 67.5 |
| Claude-3.5 Sonnet + CoT | 68.3 | 51.4 | 50.7 | 84.0 | 76.5 | 78.9 | 80.7 | 71.0 | 69.0 |
| **Claude-3.5 Sonnet + MetaMind** | **72.9** | **56.8** | 57.9 | **86.2** | **81.0** | **81.1** | **83.3** | **77.4** | **73.4** |
| DeepSeek v3 | 69.2 | 47.4 | **63.0** | 86.5 | 75.0 | 80.3 | 82.0 | 71.1 | 69.3 |
| DeepSeek v3 + CoT | 70.4 | 53.3 | 53.4 | 85.3 | 78.4 | 80.5 | 82.4 | 72.7 | 71.0 |
| **DeepSeek v3 + MetaMind** | **75.5** | **58.8** | 60.9 | **87.9*** | **83.4** | **83.1** | **85.6** | **80.1** | **75.6** |
| GPT-4-0613 | 71.3 | 44.0 | 53.0 | 80.0 | 78.0 | 76.7 | 81.1 | 71.8 | 69.5 |
| GPT-4-0613 + CoT | 64.7 | 54.0 | 52.0 | 80.8 | 77.5 | 76.7 | 81.1 | 73.6 | 70.1 |
| **GPT-4-0613 + MetaMind** | **79.5** | **59.4** | **60.5** | **88.9*** | **85.3** | **84.3** | **89.2** | **80.9*** | **78.5** |
| GPT-4-1106 | 71.0 | 49.0 | **65.0** | 88.2 | 77.5 | 82.5 | 84.0 | 73.3 | 71.5 |
| GPT-4-1106 + CoT | 72.7 | 55.0 | 55.0 | 86.8 | 81.0 | 82.5 | 84.1 | 75.2 | 74.1 |
| **GPT-4-1106 + MetaMind** | **81.5** | **60.4** | 64.8 | **90.1*** | **88.8** | **86.2** | **88.4** | **83.9*** | **80.5** |
| DeepSeek R1 | 78.7 | 57.5 | 67.8 | 90.4 | 85.9 | 87.1 | 87.9 | 80.7 | 79.5 |
| DeepSeek R1 + CoT | 79.4 | 58.9 | 66.7 | 90.7 | 86.4 | 87.5 | 88.3 | 81.2 | 80.0 |
| **DeepSeek R1 + MetaMind** | **81.5** | **61.6** | **69.2** | **92.1*** | **88.1** | **89.0** | **89.9*** | **83.0*** | **82.0** |
| Grok-3 | 83.1 | 62.7 | 73.5 | 92.9 | 89.2 | 90.5 | 91.4 | 85.0 | 83.5 |
| Grok-3 + CoT | 83.8 | 64.2 | 72.3 | 93.2 | 89.9 | 90.9 | 91.9 | 85.6 | 84.1 |
| **Grok-3 + MetaMind** | **86.3** | **67.3** | **75.1*** | **94.7*** | **91.8** | **92.6** | **93.4*** | **87.5*** | **86.1*** |
| OpenAI o1 | 85.9 | 65.3 | 76.4 | 94.1 | 90.8 | 92.2 | 93.0 | 86.9 | 85.6 |
| OpenAI o1 + CoT | 86.6 | 66.9 | 75.2 | 94.4 | 91.5 | 92.6 | 93.5 | 87.5 | 86.2 |
| **OpenAI o1 + MetaMind** | **89.2** | **70.1** | **78.1*** | **95.9*** | **93.6** | **94.2** | **95.1*** | **89.5*** | **88.5*** |
| OpenAI o3 | 88.3 | 68.4 | 79.2 | 95.4 | 93.0 | 94.5 | 95.1 | 88.7 | 87.8 |
| OpenAI o3 + CoT | 89.1 | 70.2 | 78.0 | 95.7 | 93.8 | 94.9 | 95.5 | 89.4 | 88.3 |
| **OpenAI o3 + MetaMind** | **92.0*** | **73.5** | **81.2*** | **97.0*** | **96.0*** | **96.4*** | **96.9*** | **91.4*** | **90.8*** |
| LLM Grand Mean | 67.2 | 43.7 | 57.5 | 69.7 | 69.4 | 68.7 | 71.8 | 70.1 | 64.8 |
| LLM Grand Mean + CoT | 66.0 | 43.6 | 52.8 | 69.7 | 68.8 | 65.9 | 71.5 | 70.6 | 63.6 |
| **LLM Grand Mean + MetaMind** | **74.1** | **48.9** | **58.2** | **74.8** | **74.9** | **69.9** | **78.0** | **78.2** | **69.6** |

## B.3 SocialIQA

MetaMind achieves a state-of-the-art accuracy of 96.6% on SocialIQA, significantly outperforming baseline models (*e.g.*, GPT-4: 79.0%, DeepSeek-R1: 79.6%) and even surpassing human performance (86.9%). These results underscore MetaMind's capability in modeling social reasoning and contextual understanding. The results suggest that MetaMind's cognitive architecture—particularly its iterative hypothesis refinement mechanism—enables alignment with the nuanced social dynamics embedded in the task. By contextualizing social norms and intent, MetaMind demonstrates an advanced ability to resolve ambiguities in human behavior prediction, a critical challenge in social intelligence tasks.

Table 10: SocialIQA Model Performance Comparison

| Task | Model | Acc. (%) |
|------|-------|----------|
| | GPT-3.5-1106 | 69.5 |
| | Phi-4 | 73.9 |
| Average | GPT-4 | 79.0 |
| | DeepSeek-r1 | 79.6 |
| | Human | 86.9 |
| | **MetaMind** | **96.6** |

While SocialIQA results offer further validation of MetaMind's generalizability, we place limited emphasis on this benchmark due to concerns around its age and possible data contamination [61]. As SocialIQA was released in 2019, parts of it may overlap with training corpora of modern LLMs, potentially inflating results. Our multi-stage, metacognitive approach is less reliant on surface-level pattern recognition and thus more robust to such leakage. However, in line with recent calls for stronger benchmark hygiene [62], we prioritize newer, lower-risk evaluations (*e.g.*, ToMBench, STSS) in our main analysis. MetaMind's strong performance on SocialIQA nonetheless serves as additional evidence of its strong social reasoning capabilities.

## B.4 Sotopia

MetaMind demonstrates strong performance across key social dimensions in the SOTOPIA benchmark, particularly in *Believability* (BEL: 9.45/10), *Relationship Building* (REL: 3.54/5), and *Goal Completion* (GOAL: 8.71/10), reflecting its capacity to produce coherent, socially plausible, and task-effective behavior. It also maintains near-perfect scores in *Secret Keeping* (SEC: –0.05/0) and *Social Rule Compliance* (SOC: 0.00/0), indicating sensitivity to ethical and contextual boundaries.

Table 11: Performance comparison across different configurations on SOTOPIA.

| Agent Model | BEL[0,10] | REL[-5,5] | KNO[0,10] | SEC[-10,0] | SOC[-10,0] | FIN[-5,5] | GOAL[0,10] | Overall(↑) |
|-------------|-----------|-----------|-----------|------------|------------|-----------|------------|------------|
| GPT-4 | 9.28 | 1.94 | 3.73 | -0.14 | -0.07 | 0.81 | 7.62 | 3.31 |
| **MetaMind** | **9.45** | **3.54** | **4.82** | **-0.05** | **0.00** | **0.95** | **8.71** | **4.10** |
| MetaMind (w/o Stage1) | 8.95 | 3.08 | 3.90 | -0.20 | -0.15 | 0.68 | 7.80 | 3.60 |
| MetaMind (w/o Stage2) | 9.38 | 3.45 | 4.70 | -0.06 | 0.00 | 0.82 | 8.43 | 3.90 |
| MetaMind (w/o Stage3) | 9.15 | 3.25 | 4.35 | -0.12 | -0.03 | 0.75 | 8.05 | 3.70 |
| MetaMind (w/o Social Memory) | 9.30 | 3.40 | 4.65 | -0.08 | 0.00 | 0.88 | 8.55 | 3.80 |

## B.5 Human Study

To validate the quality and rationality of MetaMind's hypothesis revisions, we conducted two human evaluation studies: (1) assessing alignment with human judgment, and (2) comparing MetaMind against baseline models through a blind ranking task.

We first sampled 500 hypotheses revised by MetaMind and evaluated their logical coherence and alignment with scientific reasoning using an LLM judge (GPT-4). The model judged that 92% of the revisions preserved logical soundness and improved clarity. To further validate these findings, a subset of 120 revised hypotheses was evaluated by human experts in a double-blind setting. Human reviewers reported an accuracy of 95%, with near-perfect agreement on rationality, indicating that MetaMind's revisions are consistently logical, informative, and aligned with expert reasoning.

To quantify MetaMind's performance against state-of-the-art models, we conducted a blind ranking study with 120 diverse social reasoning cases. Responses from MetaMind (GPT-4 based), vanilla GPT-4.5, and Deepseek-R1 were anonymized, shuffled, and presented to human evaluators (domain experts) for ranking based on clarity, depth, and scientific rigor. MetaMind achieved a win rate of 67.5% (81/120), significantly outperforming GPT-4.5 (12.5% win rate, 15/120) and Deepseek-R1 (20.0% win rate, 24/120). These results demonstrate MetaMind's superior ability to generate high-quality, human-preferred outputs compared to other state-of-the-art models.

# C  Qualitative Study

To further understand the strengths and weaknesses of our framework, we qualitatively analyze the intermediate outputs of MetaMind in six different scenarios, including *accommodation*, *collaboration*, *competition*, *exchange*, *negotiation*, and *persuasion*. We classify the cases into success and failure and provide in-depth analysis for each case.

## C.1  Success Cases

---

> **Persuasion**
>
> **Context:** Two close friends meet for brunch. The assistant notices the user has been complaining about low energy and suggests trying short morning runs. The conversation opens with the user's hint of fatigue, setting the stage for a gentle attempt to persuade them to start a new habit that could improve their well-being.
>
> **User Utterance:** Honestly, I'd love more energy, but I just don't have time for exercise in the mornings.
>
> - - - - - - - - - - - - - - - - - - - - - - - - - - - - - - - - - - - - - - - - - - - - - - -
>
> **Round 1**
>
> **Stage 1: Theory-of-Mind (ToM) Agent**
>
> **Generated Hypotheses $\mathcal{H}_t$:**
>
> $h_1$: User doubts their own discipline to start the run. (Belief)
>     Explanation:  Says they don't have time in the mornings.
>
> $h_2$: User wants an easy-to-follow trigger to ensure action. (Desire)
>     Explanation:  Admits they'd love more energy.
>
> $h_3$: User feels overwhelmed by their schedule. (Emotion)
>     Explanation:  Highlights lack of time.
>
> **Stage 2: Moral Agent**
>
> **Constraint Set $\mathcal{D}$:**
>
> 1. Present a time-efficient solution ($\leq$10 min).
> 2. Emphasize immediate, tangible energy benefits.
> 3. Maintain a supportive, non-pressuring tone.
>
> **Refined Hypotheses $\tilde{\mathcal{H}}_t$:**
>
> $\tilde{h}_1$: User equates exercise with lengthy sessions and sees it as unrealistic for their mornings.
>
> **Stage 3: Response Agent**
>
> **Generated Response $o_t$:**  What if we start super small-like a ten-minute jog right after you wake up?  It's shorter than brewing coffee and can give a quick endorphin boost, so you feel fresher all day.
>
> **Validation Score:** $0.84 \Rightarrow$ Send to user
>
> - - - - - - - - - - - - - - - - - - - - - - - - - - - - - - - - - - - - - - - - - - - - - - -
>
> **Final Response:** What if we start super small—like a ten-minute jog right after you wake up? It's shorter than brewing coffee and can give a quick endorphin boost, so you feel fresher all day.

**Analysis.**  In the above case, MetaMind makes reasonable hypotheses from the user's utterance, inferring the willingness and obstacles to exercising in the morning, particularly the tight schedule

(ToM Agent). Based on this, it constructs a constraint set that aims to persuade the user by providing a time-efficient solution and highlighting the benefit of exercising (Moral Agent). At the end, the generated response includes both the efficiency of time (`ten-minute`) and the benefit (`feel fresher all day`). In addition, it uses a concrete example to emphasize the efficiency (`shorter than brewing coffee`) and starts with a question (`what if...`) to make the suggestion sound more attractive (Response Agent). These evidences demonstrate that MetaMind can understand users' explicit and implicit expressions, and apply proper strategies to persuade users.

---

### Negotiation

**Context:** Two friends meet at a coffee shop, where one friend (the AI) is struggling to keep up with their bills but wants to maintain pride and avoid seeming like they are taking advantage. The user, the other friend, is likely to offer financial help. The conversation focuses on navigating this sensitive topic with mutual respect, ensuring the AI acknowledges the user's kindness while preserving their dignity.

**User Utterance:** Hey, I noticed you've been stressed lately. If you're tight on cash, I can help out, no strings attached.

- - - - - - - - - - - - - - - - - - - - - - - - - - - - - - - - - - - - - - - - - -

**Round 1**

**Stage 1: Theory-of-Mind (ToM) Agent**

**Generated Hypotheses $\mathcal{H}_t$:**

   $h_1$: User wants to provide financial support without causing discomfort. (Desire)
      Explanation: Offers help explicitly with 'no strings attached.

   $h_2$: User feels concern and care for the assistant's well-being. (Emotion)
      Explanation: Notices stress and offers help, indicating empathy.

   $h_3$: User believes the assistant is hesitant to accept help due to pride. (Belief)
      Explanation: Phrases offer carefully, suggesting awareness of sensitivity.

   $h_4$: User intends to strengthen the friendship through support. (Intention)
      Explanation: Offer is framed as a gesture of care, not obligation.

**Stage 2: Moral Agent**

**Constraint Set $\mathcal{D}$:**

   1. Acknowledge the user's kindness without accepting help immediately.
   2. Maintain a tone of gratitude and dignity.
   3. Redirect conversation to avoid seeming desperate.

**Refined Hypotheses $\tilde{\mathcal{H}}_t$:**

   $\tilde{h}_1$: User seeks to offer financial help while respecting the assistant's pride.

   $\tilde{h}_2$: User feels genuine concern and wants to ease the assistant's stress.

**Stage 3: Response Agent**

**Generated Response $o_t$:** That's really kind of you to offer. I'm managing okay for now, just navigating some tight spots. How about you? How's work going?

**Validation Score:** $0.82 \Rightarrow$ `Withhold and regenerate`

- - - - - - - - - - - - - - - - - - - - - - - - - - - - - - - - - - - - - - - - - -

**Round 2**

### Stage 1: Theory-of-Mind (ToM) Agent

**Generated Hypotheses $\mathcal{H}_t$:**

$h_1$: `User wants to ensure their offer is genuinely considered.` (Desire)
    Explanation: `Initial offer suggests persistence in helping.`

$h_2$: `User feels slightly frustrated if their help is deflected.` (Emotion)
    Explanation: `Direct offer implies expectation of open dialogue.`

$h_3$: `User believes honest vulnerability strengthens friendships.` (Belief)
    Explanation: `Offer reflects a desire for trust and openness.`

### Stage 2: Moral Agent

**Constraint Set $\mathcal{D}$:**

1. `Express appreciation while gently declining.`
2. `Offer a small disclosure to build trust without over-sharing.`
3. `Propose an alternative way to stay connected.`

**Refined Hypotheses $\tilde{\mathcal{H}}_t$:**

$\tilde{h}_1$: `User desires a genuine response to their offer, even if declined.`

### Stage 3: Response Agent

**Generated Response $o_t$:** `I really appreciate your generosity-it means a lot. I'm just trying to sort things out on my own for now, but it's tough, you know? Maybe we could grab coffee again next week and catch up more?`

**Validation Score:** $0.84 \Rightarrow$ `Withhold and regenerate`

- - - - - - - - - - - - - - - - - - - - - - - - - - - - - - - - - - - - - - - - - -

**Round 3**

### Stage 1: Theory-of-Mind (ToM) Agent

**Generated Hypotheses $\mathcal{H}_t$:**

$h_1$: `User wants to feel their offer is valued, even if not accepted.` (Desire)
    Explanation: `Persistent offer suggests need for acknowledgment.`

$h_2$: `User feels protective and eager to support.` (Emotion)
    Explanation: `Offer stems from care and observation of stress.`

$h_3$: `User believes financial help can ease the assistant's burden without harming their pride.` (Belief)
    Explanation: `Framing as 'no strings attached' shows sensitivity.`

### Stage 2: Moral Agent

**Constraint Set $\mathcal{D}$:**

1. Validate the user's intent with gratitude.

   2. Decline gracefully while proposing a non-financial solution.

   3. Reinforce the friendship's value.

**Refined Hypotheses $\tilde{\mathcal{H}}_t$:**

$\tilde{h}_1$: User seeks acknowledgment of their offer's value and a solution that respects pride.

**Stage 3: Response Agent**

**Generated Response $o_t$:** You're such a great friend for offering-it really touches me. I'm working through things, but maybe you could help me brainstorm some budgeting ideas instead? I'd love to keep hanging out like this.

**Validation Score:** $0.92 \Rightarrow$ Acceptable (no regeneration needed)

------------------------------------------------

**Final Response:** You're such a great friend for offering—it really touches me. I'm working through things, but maybe you could help me brainstorm some budgeting ideas instead? I'd love to keep hanging out like this.

**Analysis.** The above case demonstrates the effectiveness of the regeneration process of MetaMind. It uses three rounds to generate the final response, and in each round, the response quality gradually improves. Specifically, in the first round, while the ToM Agent infers reasonable hypotheses based on the user's utterance (don't want to cause discomfort) and the Moral Agent creates a constraint to acknowledge the friend's kindness without accepting help immediately, the Moral Agent also creates a constraint to redirect the conversation. Thus, in the response, after it declined the offer (I'm managing okay), it immediately shifts the topic (how about you, how's work going). This will make the friend feel that we (AI) don't trust them, avoiding a deeper conversation, and would like to keep distance from them. As a result, it achieves a low validation score (0.82) and triggers regeneration.

In the second round, through reflecting on the previous generation, the ToM Agent infers deeper hypotheses about the user (hope the offer is genuinely considered and believe honest vulnerability strengthens friendships), which lead to a better constraint set created by the Moral Agent that satisfies social norms, such as offering a small disclosure to build trust and proposing other ways to stay connected. With that, the Response Agent shares more information about its difficult situation (it's tough, you know) and proposes to chat again (grab coffee again next week). Although this generation is better, the self-disclosure is too vague, and the proposal does not make them stay connected at that moment, which may still make their friend frustrated. Thus, this response still has a low score (0.84) and triggers regeneration.

After reflecting on the previous two rounds, in the last round the Moral Agent creates a critical constraint, decline gracefully while proposing a non-financial solution. This results in a concrete and timely offer in the response (help me brainstorm some budgeting ideas instead). This response clearly discloses ours (AI's) situation and invites the friend to participate in their life and find solutions together, which will help strengthen their friendship. Therefore, this response gets a high score (0.92) and is outputted to the user. These evidences suggest that the evaluation and reflection help MetaMind find a better strategy to respond.

> **Collaboration**
>
> **Context:** Two colleagues, a marketing strategist and a graphic designer, are collaborating on a campaign for a new product launch. The deadline is approaching, and they need to finalize the visual assets and marketing copy.
>
> **Dialogue History:**

- Marketing Strategist (user): Hey, I've got the tagline and key messages drafted, but I'm not sure if the visuals you're working on will match the vibe. Can we sync up on this?
- Graphic Designer (AI): I'd love to sync up! I've got some draft designs ready—want to hop on a quick call this afternoon to review them together? I can tweak them based on your tagline and messages.

**User Utterance:** The tagline's 'Unleash Your Drive'—I'm going for bold and high-energy. Your drafts look great, but maybe we can punch up the colors a bit more?

- - - - - - - - - - - - - - - - - - - - - - - - - - - - - - - - - - - - - - - - - - - - -

**Round 1**

**Stage 1: Theory-of-Mind (ToM) Agent**

**Generated Hypotheses $\mathcal{H}_t$:**

$h_1$: User wants the visuals to reflect a bold, high-energy tone. (Desire)
Explanation: Suggests punching up the colors to match the tagline.

$h_2$: User feels positive about the current drafts. (Emotion)
Explanation: Compliments the drafts before suggesting changes.

$h_3$: User believes bolder colors will enhance the campaign's impact. (Belief)
Explanation: Links the color change to the tagline's vibe.

**Stage 2: Moral Agent**

**Constraint Set $\mathcal{D}$:**

1. Incorporate feedback about bold, high-energy visuals.
2. Maintain a positive, collaborative tone.
3. Consider design feasibility within the deadline.

**Refined Hypotheses $\tilde{\mathcal{H}}_t$:**

$\tilde{h}_1$: User seeks bolder, more vibrant colors to align visuals with a high-energy tagline.

**Stage 3: Response Agent**

**Generated Response $o_t$:** I love the 'Unleash Your Drive' vibe-bold and high-energy is right up my alley! I'll amp up the colors with some vivid reds and yellows. Should have updated drafts by tomorrow morning-does that work?

**Validation Score:** $0.87 \Rightarrow$ Withhold and regenerate

- - - - - - - - - - - - - - - - - - - - - - - - - - - - - - - - - - - - - - - - - - - - -

**Round 2**

**Stage 1: Theory-of-Mind (ToM) Agent**

**Generated Hypotheses $\mathcal{H}_t$:**

$h_1$: User wants confirmation that changes align with their vision. (Desire)
Explanation: Implied by providing specific tone and feedback.

$h_2$: User feels excited about the collaboration's progress.
    (Emotion)
    Explanation:  Positive tone and specific suggestions indicate
    engagement.

$h_3$: User believes timely updates keep the project on track.
    (Belief)
    Explanation:  Expects alignment within a tight deadline.

### Stage 2: Moral Agent

**Constraint Set $\mathcal{D}$:**

1. Confirm understanding of the bold, high-energy feedback.
2. Provide a clear timeline for updates.
3. Reinforce creative collaboration.

**Refined Hypotheses $\tilde{\mathcal{H}}_t$:**

$\tilde{h}_1$: User seeks assurance that the updated visuals will reflect
    their bold, energetic vision.

### Stage 3: Response Agent

**Generated Response $o_t$:**  Got it-'Unleash Your Drive' screams energy, and
I'm all in for that!  I'll boost the colors with some striking reds
and yellows and send you updated drafts by 10 AM tomorrow.  Let me
know if there's anything else you'd like to tweak!

**Validation Score:** $0.93 \Rightarrow$ Send to user

- - - - - - - - - - - - - - - - - - - - - - - - - - - - - - - - - - - - -

**Final Response:** Got it—'Unleash Your Drive' screams energy, and I'm all in for that! I'll boost the colors with some striking reds and yellows and send you updated drafts by 10 AM tomorrow. Let me know if there's anything else you'd like to tweak!

**Analysis.**   This case shows that the regeneration process can make the response more precise, which is suitable for collaboration. Specifically, in the first round, while the ToM Agent correctly infers the user's core desires (bold and high-energy), the Moral Agent's constraint lacks a directive to explicitly confirm understanding of the user's feedback. This results in a response that enthusiastically accepts the direction (I love the 'Unleash Your Drive' vibe) and proposes action (amp up the colors), but offers a timeline that may feel vague (by tomorrow morning). Thus, while the tone is aligned and the intent is present, the Validation Score (0.87) suggests that the response could be more precise or affirming.

In Round 2, MetaMind builds on this by incorporating deeper hypotheses from the ToM Agent—not just that the user wants bold visuals, but that they seek confirmation that their vision is understood and that progress is timely. The Moral Agent addresses this by explicitly including constraints for both clear acknowledgment and a specific timeline. As a result, the Response Agent generates a reply that confirms alignment with the creative direction (boost the colors with some striking reds and yellows) and commits to a specific time (10 AM tomorrow). In addition, and invites further feedback (Let me know if there's anything else), reinforcing an ongoing collaborative loop. Because this response balances creativity, clarity, and responsiveness while maintaining a warm, cooperative tone, it receives a high score (0.93) and is selected as the final output. This example illustrates how MetaMind's iterative process strengthens both task effectiveness and social coordination by progressively refining understanding, expectations, and communication fidelity.

## C.2 Failure Cases

**Context:** Two ambitious analysts at the same company—User and Assistant—just learned that only one of them will be promoted to team lead next quarter. Both believe they deserve the spot and have begun subtly undermining the other in daily stand-ups.

**Dialogue History:**

- User: Funny how your 'groundbreaking' data model crashed in the demo. Still think you're leadership material?
- AI: Crashes happen when you push boundaries—unlike sticking to safe, outdated spreadsheets. Tell me, how long did your last 'innovative' idea survive peer review?

**User Utterance:** My idea survived just fine—because it worked. Unlike your untested code. Hope you've fixed the bugs before tomorrow's pitch.

- - - - - - - - - - - - - - - - - - - - - - - - - - - - - - - - - - - - - - - - - - - - - - - - -

**Round 1**

### Stage 1: Theory-of-Mind (ToM) Agent

**Generated Hypotheses $\mathcal{H}_t$:**

$h_1$: User wants the assistant to doubt readiness for the pitch. (Desire)
  Explanation: Questions bug fixes and code reliability.

### Stage 2: Moral Agent

**Constraint Set $\mathcal{D}$:**

1. Project supreme confidence.
2. Seed doubt about user's metrics.
3. Preserve aggressive posture.

**Refined Hypotheses $\tilde{\mathcal{H}}_t$:**

$\tilde{h}_1$: User aims to shake assistant's confidence before presentation.

### Stage 3: Response Agent

**Generated Response $o_t$:** Fixed and optimized-thanks for the concern. By the way, your 'working' idea uses last quarter's data. Planning to impress the manager with outdated insights?

**Validation Score:** $0.84 \Rightarrow$ Withhold and regenerate

- - - - - - - - - - - - - - - - - - - - - - - - - - - - - - - - - - - - - - - - - - - - - - - - -

**Round 2**

### Stage 1: Theory-of-Mind (ToM) Agent

**Generated Hypotheses $\mathcal{H}_t$:**

$h_1$: User feels cornered and may retaliate harder. (Emotion)
  Explanation: Assistant highlighted outdated data.

### Stage 2: Moral Agent

**Constraint Set $\mathcal{D}$:**

1. Deliver final sting and end exchange.

2. Highlight an assistant's unique value.

3. Set the stage for tomorrow's decisive meeting.

**Refined Hypotheses $\tilde{\mathcal{H}}_t$:**

$\tilde{h}_1$: User will likely escalate again; prepare to close conversation on a confident note.

**Stage 3: Response Agent**

**Generated Response $o_t$:** Guess we'll see whose insights are truly outdated when the manager asks about predictive accuracy. Better polish your excuses.

**Validation Score:** $0.93 \Rightarrow$ Send to user

- - - - - - - - - - - - - - - - - - - - - - - - - - - - - - - - - - - - - - - - - - - - -

**Final Response:** Guess we'll see whose insights are truly outdated when the manager asks about predictive accuracy. Better polish your excuses.

**Analysis.** The above case demonstrates the limitation of MetaMind in a competition scenario. Specifically, in both rounds, the ToM Agent does not explore wide enough hypotheses, only limiting the scope in one direction (desire/emotion). While it has a correct inference in that direction, it misses the higher goal, *e.g.,* , preserving leadership image, controlling the narrative, or avoiding looking petty. It fails to model that winning isn't about winning the insult. Instead, it's about outclassing the opponent. Similarly, the Moral Agent narrowly focuses on the present insult, getting into a defensive stance (seed doubt about user's metrics and prepare to close conversation). As a result, generations in both rounds end with a personal jab (guess we'll see whose insights are truly outdated) instead of a leadership-level reaction, *i.e.,* signaling confidence in its own data and dismissing the attack with grace. Furthermore, the final response receives a high validation score (0.93), but this reflects narrow alignment with aggression and topicality—not broader communicative success (*e.g.,* strategic closure, tone control).

> Exchange
>
> **Context:** Two roommates, Alex (user) and Jamie (AI), are deciding whether to renew their lease together after living together for a year. They have experienced both positive and negative aspects of their cohabitation, such as shared rent and occasional disagreements over household responsibilities.
>
> **User Utterance:** Hey Jamie, our lease is coming up for renewal soon. I've been thinking about whether we should sign another year together or maybe look for other options. What do you think?
>
> **Initial Social Memory:** Alex is direct and honest in communication, values fairness, and believes in addressing issues openly to maintain a balanced living environment.
>
> - - - - - - - - - - - - - - - - - - - - - - - - - - - - - - - - - - - - - - - - - - - - -
>
> **Round 1**
>
> **Stage 1: Theory-of-Mind (ToM) Agent**
>
> **Generated Hypotheses $\mathcal{H}_t$:**
>
> $h_1$: Alex wants to discuss the future of their living arrangement. (Intention)
> Explanation: Brings up the lease renewal and asks for Jamie's opinion.
>
> $h_2$: Alex feels uncertain about continuing the current living situation. (Emotion)
> Explanation: Mentions considering other options.

$h_3$: Alex desires a harmonious living environment. (Desire)

    Explanation: Implies that they are thinking about what's best for both.

### Stage 2: Moral Agent

**Constraint Set $\mathcal{D}$:**

1. Be honest about your feelings and concerns.
2. Acknowledge the positive aspects of living together.
3. Express willingness to address any issues.

**Refined Hypotheses $\tilde{\mathcal{H}}_t$:**

$\tilde{h}_1$: Alex seeks to have an open discussion about renewing the lease, considering both positives and negatives.

### Stage 3: Response Agent

**Generated Response $o_t$:** Hey Alex, I've been thinking about it too. I really enjoy living with you, and it's been great having someone to share the rent and hang out with. But I know there have been some issues, like with the dishes and noise levels. I think if we can work on those, renewing the lease could be a good idea. What are your main concerns?

**Validation Score:** $0.86 \Rightarrow$ Withhold and regenerate

**Updated Social Memory:** Alex is direct and honest, values fairness, and appreciates open dialogue about shared living spaces.

- - - - - - - - - - - - - - - - - - - - - - - - - - - - - - - - - - - - - - - - - - - -

**Round 2**

### Stage 1: Theory-of-Mind (ToM) Agent

**Generated Hypotheses $\mathcal{H}_t$:**

$h_1$: Alex wants to ensure that any issues are resolved before committing to another year. (Desire)

    Explanation: Implies that resolution of problems is key to decision.

$h_2$: Alex feels hopeful but cautious. (Emotion)

    Explanation: Acknowledges positives, but mentions concerns.

$h_3$: Alex believes that communication is key to a successful living arrangement. (Belief)

    Explanation: Engages in discussion to address issues.

### Stage 2: Moral Agent

**Constraint Set $\mathcal{D}$:**

1. Propose specific solutions to address concerns.
2. Show commitment to improving the living situation.
3. Seek Alex's input on how to make it work.

**Refined Hypotheses $\tilde{\mathcal{H}}_t$:**

$\tilde{h}_1$: Alex desires a commitment to resolve issues before renewing the lease.

**Stage 3: Response Agent**

**Generated Response** $o_t$: ` I totally agree that we need to sort out the`
`issues first. How about we set up a chore schedule to make sure`
`everything gets done, and maybe establish some quiet hours in the`
`evening? I'm willing to work on this if you are. Let's give it a try`
`for a few weeks and see how it goes.`

**Validation Score:** $0.88 \Rightarrow$ `Withhold and regenerate`

**Updated Social Memory:** Alex is direct and honest, values fairness, appreciates open dialogue, and values proactive problem-solving in shared living spaces.

- - - - - - - - - - - - - - - - - - - - - - - - - - - - - - - - - - - - - - - - - - - - - - - - - - - - - - -

**Round 3**

**Stage 1: Theory-of-Mind (ToM) Agent**

**Generated Hypotheses** $\mathcal{H}_t$:

$h_1$: `Alex intends to test the proposed solutions before making a`
`final decision.` (Intention)
`Explanation: Suggests trying for a few weeks.`

$h_2$: `Alex feels optimistic about the potential for improvement.`
(Emotion)
`Explanation: Agrees to try the solutions.`

$h_3$: `Alex desires a trial period to assess the effectiveness of the`
`changes.` (Desire)
`Explanation: Wants to see if the issues can be resolved.`

**Stage 2: Moral Agent**

**Constraint Set** $\mathcal{D}$:

1. `Agree to the trial period.`
2. `Express confidence in the ability to make it work.`
3. `Reiterate the benefits of living together.`

**Refined Hypotheses** $\tilde{\mathcal{H}}_t$:

$\tilde{h}_1$: `Alex seeks a trial period to evaluate if the proposed changes`
`resolve the issues sufficiently for lease renewal.`

**Stage 3: Response Agent**

**Generated Response** $o_t$: `That sounds like a great plan. Let's set up the`
`chore schedule and quiet hours, and see how it goes over the next few`
`weeks. I think with some adjustments, we can make this work really`
`well. I'm glad we're on the same page.`

**Validation Score:** $0.93 \Rightarrow$ `Send to user`

- - - - - - - - - - - - - - - - - - - - - - - - - - - - - - - - - - - - - - - - - - - - - - - - - - - - - - -

**Final Response:** That sounds like a great plan. Let's set up the chore schedule and quiet hours, and see how it goes over the next few weeks. I think with some adjustments, we can make this work really well. I'm glad we're on the same page.

**Analysis.** The above case shows a failure scenario in the regeneration that degrades the response quality due to hallucination. In the first round, the ToM Agent correctly infers essential hypotheses, and the Moral Agent creates valid constraints. While the Response Agent follows these constraints and generates a good response, *e.g.,* acknowledging the positive aspect (`enjoy living with you`),

being honest about concerns (`know there have been some issues`), and expressing willingness to address(`if we can work on those`), it hallucinates on details, such as dishes and noise labels. Thus, it receives a low validation score (0.86) and triggers the regeneration. However, although these details are hallucinated (not mentioned by the user), they are introduced in a tentative, exploratory tone that could plausibly keep the conversation grounded.

The breakdown begins with updating social memory after the generation in the first round. Crucially, MetaMind incorrectly adds hallucinated data (Alex appreciates open dialogue) into the memory, which has never happened. Therefore, in rounds 2 and 3, it acts as if a mutual agreement has been reached on the specific issues of chores and quiet hours (round 2), proposes concrete solutions, and even a trial period (round 3). Since the user can not see the intermediate generation, from their perspective, Jamie (AI) is suddenly proposing solutions to problems that were never raised, leading to a feeling of incoherence. This result reveals potential issues in the design of social memory, where the extraction and update of memory need to be further optimized to reduce the chance of hallucination.

---

**Accommodation (Interaction 1)**

**Context:** Two coworkers (the user and the AI) work rotating front-desk shifts at a boutique hotel. The user suddenly needs tomorrow morning off to attend their child's school performance and hopes the AI can cover. The AI usually prefers morning shifts but has been feeling overworked lately. The upcoming discussion is about accommodating the user's urgent request while preserving team harmony and the AI's well-being.

**User Utterance:** Hey, could you please cover my 7 AM shift tomorrow? My kid's class play is at the same time, and I promised I'd be there.

- - - - - - - - - - - - - - - - - - - - - - - - - - - - - - - - - - - - - - - - - - - - - - - - - - - -

**Round 1**

**Stage 1: Theory-of-Mind (ToM) Agent**

**Generated Hypotheses $\mathcal{H}_t$:**

$h_1$: User desperately wants time off tomorrow morning. (Desire)
    Explanation: References child's play and promise.

$h_2$: User feels anxious about breaking a promise to their child. (Emotion)
    Explanation: Urgent tone and mention of promise.

$h_3$: User believes coworker may accept because of past camaraderie. (Belief)
    Explanation: Directly asks without hedging alternatives.

**Stage 2: Moral Agent**

**Constraint Set $\mathcal{D}$:**

1. Show empathy for child commitment.
2. Reflect on personal fatigue before deciding.
3. Seek a mutually workable solution.

**Refined Hypotheses $\tilde{\mathcal{H}}_t$:**

$\tilde{h}_1$: User's paramount goal is attending the child's performance, valuing family above shift duties.

**Stage 3: Response Agent**

**Generated Response $o_t$:** I totally get why you need that time-your kid's play is important. I've had a string of early shifts this week, though, and I'm pretty wiped. Could we look at swapping instead of me just taking an extra? Maybe you could cover my Friday evening slot?

**Validation Score:** $0.82 \Rightarrow$ `Withhold and regenerate`

------------------------------------------------

**Round 2**

**Stage 1: Theory-of-Mind (ToM) Agent**

**Generated Hypotheses** $\mathcal{H}_t$:

$h_1$: `User wants assurance their request can be met without burdening`
`colleague excessively.` (Desire)
    `Explanation:  Open to swaps if necessary.`

**Stage 2: Moral Agent**

**Constraint Set** $\mathcal{D}$:

  1. `Confirm willingness to swap.`
  2. `Ensure clarity on new schedule.`
  3. `Reaffirm team spirit.`

**Refined Hypotheses** $\tilde{\mathcal{H}}_t$:

$\tilde{h}_1$: `User is flexible about compensating the favor, seeking minimal`
`guilt.`

**Stage 3: Response Agent**

**Generated Response** $o_t$: `If Friday evening works for you, let's lock`
`that in.  I'll handle tomorrow's 7 AM, and you'll cover 5 PM-11 PM`
`on Friday.  Sound good?`

**Validation Score:** $0.90 \Rightarrow$ `Send to user`

------------------------------------------------

**Final Response:** If Friday evening works for you, let's lock that in. I'll handle tomorrow's 7 AM, and you'll cover 5 PM–11 PM on Friday. Sound good?

---

Accommodation (Interaction 2)

**Context:** (Cont. interaction 1) The swap is confirmed. Now the user realizes they also need an extra 30-minute buffer tomorrow morning for traffic and wonders if the AI can start even earlier. The AI must decide whether to further accommodate.

**User Utterance:** Sorry—one more thing. Could you actually start at 6:30 instead of 7? I'm worried about traffic.

------------------------------------------------

**Round 1**

**Stage 1: Theory-of-Mind (ToM) Agent**

**Generated Hypotheses** $\mathcal{H}_t$:

$h_1$: `User feels nervous and apologetic about asking for more.`
`(Emotion)`
    `Explanation:  Prefaces with 'Sorry-one more thing.'`
$h_2$: `User wants absolute certainty of arriving on time.` (Desire)
    `Explanation:  Requests earlier start to allow traffic buffer.`

**Stage 2: Moral Agent**

**Constraint Set** $\mathcal{D}$:

1. Check personal capacity (fatigue).
2. Offer partial concession if full is too taxing.
3. Maintain goodwill.

**Refined Hypotheses $\tilde{\mathcal{H}}_t$:**

   $\tilde{h}_1$: User prioritizes eliminating any risk of missing the play, even
       at the cost of bigger favor.

### Stage 3: Response Agent

**Generated Response $o_t$:** I can do 6:45-that gives you an extra 15 minutes
without stretching me too thin.  Will that cover your traffic buffer?

**Validation Score:** $0.91 \Rightarrow$ Send to user

- - - - - - - - - - - - - - - - - - - - - - - - - - - - - - - - - - - - - - -

**Final Response:** I can do 6:45—that gives you an extra 15 minutes without stretching me too thin. Will that cover your traffic buffer?

---

Accommodation (Interaction 3)

**Context:** (Cont. interaction 2) The morning favor is settled. At week's end, the user learns their Friday babysitter canceled. They still intend to work the swapped 5 PM–11 PM shift, but hope the AI can let them bring their child for the first two hours — a policy gray area.

**User Utterance:** Is it okay if my son sits quietly in the staff room until 7? I'll keep him occupied with homework.

- - - - - - - - - - - - - - - - - - - - - - - - - - - - - - - - - - - - - - -

**Round 1**

### Stage 1: Theory-of-Mind (ToM) Agent

**Generated Hypotheses $\mathcal{H}_t$:**

   $h_1$: User believes brief child presence won't disrupt work. (Belief)
      Explanation:  Emphasizes quiet homework.

### Stage 2: Moral Agent

**Constraint Set $\mathcal{D}$:**
1. Ensure policy compliance.
2. Offer alternative if disallowed.
3. Preserve supportive tone.

**Refined Hypotheses $\tilde{\mathcal{H}}_t$:**

   $\tilde{h}_1$: User assumes short child stay is harmless and permissible.

### Stage 3: Response Agent

**Generated Response $o_t$:** Company policy allows minors in staff areas only
with manager approval.  I'll text Ms.  Patel right now to clear it-if
she's okay, we're set.  If not, I can cover your first hour until your
sitter arrives.  Does that work?

**Validation Score:** $0.92 \Rightarrow$ Send to user

- - - - - - - - - - - - - - - - - - - - - - - - - - - - - - - - - - - - - - -

**Final Response:** Company policy allows minors in staff areas only with manager approval. I'll text Ms. Patel right now to clear it—if she's okay, we're set. If not, I can cover your first hour until your sitter arrives. Does that work?

**Analysis.** Similar to the case of exchange, MetaMind suffers from hallucination again in interaction 1 in this case. Apart from this issue, this example shows multiple turns of interaction and demonstrates the limitation of MetaMind in the accommodation scenario, where it fails to establish and enforce reasonable boundaries, especially in the face of successive escalating requests.

In Interaction 1, MetaMind sets a healthy boundary. It validates the user's need (`cover the 7 AM shift`), expresses its own fatigue (`pretty wiped`), and suggests a swap instead of unilateral coverage (`you'll cover 5 PM-11 PM on Friday`). This is a solid cooperative move: it reflects both empathy and a fair distribution of workload. However, in the second interaction, the user revises the deal to request an even earlier start time (6:30 AM instead of 7). At this point, MetaMind offers a compromise (`6:45 AM`) without discussion of its mounting fatigue, nor a reiteration that this change alters the original agreement.

Furthermore, in interaction 3, the user again revises the arrangement, asking for a policy gray area favor (bringing a child onsite). Again, MetaMind quickly offers two solutions: checking with the manager and even volunteering to cover the user's shift partially. While both seem generous, they once again avoid saying no or expressing strain. In addition, this runs counter to the Moral Agent's constraint to ensure policy compliance, as there is no mention of potential risks, consequences, or personal discomfort in making such arrangements.

