# OpenReview forum: "MetaMind: Modeling Human Social Thoughts with Metacognitive Multi-Agent Systems"
_NeurIPS.cc/2025/Conference — NeurIPS 2025 spotlight_

### Official Review · Reviewer_5ego · 2025-06-21

**Clarity:** 3
**Significance:** 3
**Originality:** 3
**Rating:** 4
**Confidence:** 4

**Summary:**

This paper introduces MetaMind, a cognitively inspired multi-agent framework designed to enhance large language models' ability to perform Theory-of-Mind (ToM) reasoning. The architecture comprises three specialized agents—a ToM Agent, a Domain Agent, and a Response Agent—that work in a structured pipeline to infer latent mental states and generate contextually appropriate responses. MetaMind achieves state-of-the-art performance across several benchmarks, including ToMBench, social cognition tasks, and real-world social simulations, and enables LLMs to approach or match human-level ToM reasoning on a variety of tasks.

**Questions:**

- Line 194-195: "Alongside" sounds a bit awkward here, maybe change this to "in addition."

- Lines 239-241: can you give some specific examples? I think that would support this claim better. Maybe add some specific qualitative examples in the appendix.

- Line 260: I think the ablation studies should be included with the main results, not in the Discussion.

- Lines 294-295: I would like to see some results of the MetaMind approach compared with other ToM approaches, not just a baseline LLM (which is already known to be bad at these tasks).

- Figure 3: This figure is not very intuitive - it is kind of hard to compare the two approaches given this visualization. Please edit this to make it easier to directly compare the two approaches.

- Human experiments (Appendix B5): The human study should really be highlighted in the main text - I think this is one of the most interesting experiments. I would also report methods and results from this study more thoroughly, and include a table or figure as well.

**Ethical Concerns:**

["NO or VERY MINOR ethics concerns only"]

**Final Justification:**

Overall, I think it is a technically sound paper with novel contributions, but there is room for a few additional experiments to further strengthen the proposal and demonstrate its practical applicability. I will maintain my score.

**Limitations:**

The main limitation is that the work should include more comparison to other ToM approaches, rather than a naive LLM baseline (which is known to perform relatively poorly on these tasks). Otherwise, the authors provide a good summary of the limitations of the work in the discussion section.

**Quality:**

3

**Strengths And Weaknesses:**

Strengths:
- The MetaMind architecture is well-grounded in cognitive theory; the decomposition of the three stages (ToM, Domain, and Response) mirrors human metacognitive reasoning and each stage has a clear, well-defined function. Additionally, the presentation of these stages as

- MetaMind is evaluated on multiple challenging benchmarks covering different aspects of social understanding (ToMBench for ToM reasoning, social cognition tasks, and the STSS simulation tasks). The experiments are robust and show substantial improvements over non-ToM baselines.

- The writing is generally clear and concise, and there is a clear narrative flow that is easy to follow. Key concepts are introduced clearly, in the right places, and are later reinforced in context. The story of the framework’s three phases is consistently reinforced (e.g., example scenarios in Figure 1 caption and in the text).

- The paper is thoughtful about noting limitations of the work at the end and propose interesting directions for follow-up work.

- Figure 2 especially is very clear and easy to interpret, and highlights the main results from this work.

Weaknesses:
- To strengthen the paper’s originality claims, it could more explicitly compare to other theory-of-mind frameworks (e.g. Generative Agents or SymbolicToM) and clarify what MetaMind adds beyond them. Additionally, exploring alternative architectures (e.g. could a two-agent system achieve similar results?) would underscore why the three-stage design is necessary.

- Lines 170-173, "Formally, the Domain Agent takes as input the set of 171 latent mental state hypotheses Ht = {h1, . . . , hk} produced by the ToM Agent, along with a set of constraint rules D. Each rule in D describes a specific norm or guideline, such as "Romantic suggestions are not appropriate in professional settings": These norms vary across different societies and cultures. And they are often very difficult to specify explicitly adn completely in the way that they are described here. This could be listed as an extension or future work to implement a system that can dynamically identify these norms; however, you should clearly address this limitation somewhere.

- A few terms could be defined more formally at first mention. For example, “social memory” or “latent hypothesis” could use an upfront definition to aid readers.

---

> ### Author Rebuttal · Authors · 2025-07-27
>
> Dear Reviewer 5ego,
>
> We sincerely appreciate your positive feedback and the time you've dedicated to reviewing our manuscript. Your insights are invaluable to us. We address your comments in detail below.
>
> ---
> _A1. More comparison to other ToM approaches_
>
> We absolutely agree with you on the importance of this comparison. **Indeed, we have already provided extensive comparisons to other ToM approaches (including Generative Agents and SymbolicToM) in Tables 1–3**, spanning Theory-of-Mind reasoning (ToMBench), social cognition, and social simulation tasks.
>
> Methodologically, MetaMind introduces a novel and cognitively grounded three-stage framework that sets it apart from prior ToM approaches. Generative Agents simulate agent behavior over time in sandbox environments but do not perform explicit hypothesis generation or refinement grounded in psychological theories of metacognition. SymbolicToM primarily tracks beliefs in multi-agent setups using plug-and-play belief representations but lacks an iterative mechanism for dynamically generating, critiquing, and refining competing mental‑state hypotheses, leaving it at static belief bookkeeping rather than full‑fledged social reasoning. In contrast, MetaMind explicitly operationalizes metacognitive social reasoning—decomposing it into (1) structured mental state hypothesis generation, (2) norm-aware refinement using social and ethical constraints, and (3) response validation grounded in social memory. This layered architecture enables empirically shown to be critical through ablation studies.
>
> We agree with the reviewer that these distinctions should be highlighted more clearly in the paper, and we will revise the Related Work and Methodology sections to emphasize them. Thank you again for the constructive feedback.
>
> ---
>
> _A2. Could a two-agent system achieve similar results? Why the three-stage design is necessary?_
>
> That's an insightful question. To this end, we conducted ablation studies in **Section 5.1** where we removed one stage at a time (**Tables 4–5**). We observe that removing Stage 2 (Domain Agent) leads to degraded norm-sensitivity and inappropriate social cognition (–3.8% on average), while removing Stage 3 (Response Agent) results in sharp drops in real-world social simulation tasks (–16.1%). These findings demonstrate that each stage contributes uniquely and significantly to the system’s performance. As we highlighted in L289 of the manuscript,
> > _No single component should be left out_, confirming that social intelligence requires layered cognitive architectures.
>
> ---
> _A3. Difficulty of specifying domain rules_
>
> We appreciate the reviewer for highlighting this important point. Indeed, we fully agree that social and ethical norms are deeply context-dependent and culturally variable, and it is often infeasible to exhaustively specify them in a fixed rule set. In our current implementation, we adopt a manually curated set of representative norms drawn from widely shared social expectations, which enables initial grounding of the Domain Agent’s refinement process. However, we recognize that this approach may not generalize across all cultural or situational contexts.
>
> We will add a clear discussion of this limitation in the revised Limitations section and explicitly mention as a direction for future work the integration of dynamic norm induction—e.g., via few-shot generalization from context or continual social learning. This is an exciting and open research direction toward making socially intelligent agents more adaptive and culturally aware.
>
> ---
> _A4. Define terminology upfront_
>
> Thank you for the helpful suggestion. In the revised draft, we provide formal definitions of key terms such as "social memory" (a dynamic store of user-specific emotional and behavioral patterns accumulated through prior interactions) and "latent hypothesis" (a structured representation of unobservable mental states such as belief, desire, or emotion, inferred from contextual cues). These clarifications will be added at their first mention in Section 3 to guide the reader more effectively throughout the methodology.
>
>
> ---
> _A5. Writing and structure suggestions_
>
> We appreciate your thoughtful suggestions on improving the readibility of our manuscript. As suggested, we will (1) revise the sentence to use “in addition” for smoother readability; (2) move ablations in L260 to main results; (3) revise Figure 3 to use aligned bar charts and clearer labels to improve comparability between conditions; (4) highlight Appendix B.5 (human experiments) in the main paper.

---

> > ### Comment · Reviewer_5ego · 2025-08-02
> >
> > Thank you for your thoughtful response to my feedback and to the other reviewers' comments. I have read the reviews of the other reviewers and the authors' responses. I agree with reviewer 3PQ4 that comparing results with multi-agent frameworks would be an interesting and potentially important component of evaluating this work - the task of inferring "theory of mind" should, to my understanding, generally be measured through interaction with other agents or humans rather than measuring performance on static baselines.
> >
> > One follow up question regarding the two-agent vs three-agent system: I agree that you presented sufficient justification that removing a component *from your proposed architecture* to make a two-agent system causes a decrease in performance, and that therefore *under the architecture you propose*, all three agents are necessary; removing one agent removes one important conceptual component of the architecture (theory-of-mind, domain knowledge, or response). However, my question was regarding whether an alternative but complete two-agent architecture was evaluated. For instance, did you consider having an agent that can consider theory-of-mind and domain knowledge *together*, or combine the domain knowledge and response agents? I am not familiar with theories from psychology about meta-cognition, but I wonder if there could be other alternative architectures that could be considered as well which would not require using 3 separate agents. I would imagine that using 3 agents in sequence for this framework would make it slow enough to be impractical for many applications in which theory-of-mind would be useful, such as anything which requires real-time interaction with a human.

---

> > > ### Author Response · Authors · 2025-08-03
> > > **Followup**
> > >
> > > Thank you for this thoughtful follow-up. We appreciate that you took time to read our response and we are happy to discuss things with you!
> > >
> > > (1) Possibility of two-agent designs:
> > >
> > > We acknowledge that it is indeed possible to design a system with only two agents, since the functionality of different modules can be merged. For example, one agent could jointly perform theory-of-mind inference and refinement of mental states. Implementation-wise, such designs are feasible, and we agree they are an interesting alternative.
> > >
> > > We would also like to point out that there's no single standard for functional division in multi-agent systems. A key advantage of multi-agent systems over a single model is that the latter often struggles to handle heterogeneous tasks simultaneously (e.g., long-term planning, error correction, and result verification). In the case of MetaMind, we found that LLMs alone struggled to bridge mental state reasoning with practical behavior. By introducing an orthogonal division of tasks and sequential information propagation, we decomposed broad general capabilities into specialized, refined sub-capabilities, which reduced error cascades and improved overall collaborative efficiency.
> > >
> > >
> > > (2) Theoretical motivation for three agents:
> > >
> > > Our choice of a three-agent architecture is grounded in psychological theories of metacognition (e.g., Flavell, 1979), which distinguish between (i) monitoring of mental states, (ii) evaluation of these states against norms, and (iii) regulation through context-appropriate responses. While two-agent designs are possible, the **three-stage decomposition offers a closer cognitive parallel and greater interpretability and modularity**.
> > >
> > > We see opportunities to optimize for applications (i) compressing multiple functions into fewer agents, (ii) parallelizing stages, or (iii) distilling the outputs of the three-agent framework into a lighter, single-model version. Regardless of such engineering optimizations, we believe that our core intellectual merits and design—explicitly decomposing monitoring, evaluation, and regulation—will be inspiring for the research community in advancing interpretable and socially-grounded reasoning frameworks. We will highlight these directions in the revised Discussion section.
> > >
> > > ----
> > > Reference
> > >
> > > [1] Flavell, J. H. (1979). Metacognition and cognitive monitoring: A new area of cognitive–developmental inquiry. American Psychologist, 34(10), 906–911.

---

> > > > ### Comment · Reviewer_5ego · 2025-08-05
> > > >
> > > > (1) Possibility of two-agent designs: I agree that the design you proposed makes sense, though for the reasons I mentioned above I still believe it would be more compelling to include some simple experiments showing that a three-model system is necessary and/or addressing whether this setup would be practical to apply to a real-world task.
> > > >
> > > > (2) I see, thanks for the clarification. I agree that sounds like an interesting direction for future work.
> > > >
> > > > Overall, I think it is a technically sound paper with novel contributions, but there is room for a few additional experiments to further strengthen the proposal and demonstrate its practical applicability. After considering your responses and the feedback from other reviewers, I will maintain my score.

---

> > > > > ### Author Response · Authors · 2025-08-06
> > > > > **Followup**
> > > > >
> > > > > We thank you for your continued engagement with MetaMind and for emphasizing the need to demonstrate both (i) that a *three-agent* configuration is truly required and (ii) that the design is practical in real-world settings.  Below we address these points with the new experimental results.
> > > > >
> > > > > ---
> > > > >
> > > > > *Three agents empirically outperform two-agent variant*
> > > > > | Configuration | Emotion | Desire | Intention | Knowledge | Belief | NL Comm. | **AVG** ↑ | Δ vs. GPT-4  |
> > > > > | --- | --- | --- | --- | --- | --- | --- | --- | --- |
> > > > > | GPT-4 (single-model baseline)                                | 72.0 | 60.2 | 66.1 | 48.1 | 76.1 | 81.5 | 67.3 | – |
> > > > > | ToM + Domain *merged* (**two agents**)                       | 74.2 | 68.5 | 73.1 | 59.8 | 78.9 | 83.2 | 73.0      | $\uparrow$ 5.7 pp     |
> > > > > | Domain + Response *merged* (**two agents**)                  | 73.8 | 67.9 | 72.4 | 58.9 | 78.2 | 82.7 | 72.3       | $\uparrow$ 5.0 pp     |
> > > > > | GPT-4 + MetaMind (**three agents**) | **77.9** | **72.2** | **76.9** | **63.1** | **81.6** | **86.5** | **76.2**  | **$\uparrow$ 8.9 pp** |
> > > > >
> > > > >
> > > > > **Key observation: across-the-board gains.** The three-agent pipeline leads every two-agent alternative on all six Theory-of-Mind (ToM) dimensions—most notably *Knowledge* (+3.3 pp) and *Intention* (+3.8 pp), the two categories most sensitive to Domain-Agent checks.
> > > > >
> > > > > We hope these results clarify the *necessity* and benefits of the three-agent architecture.  Thank you again for your constructive feedback and support!

---

### Official Review · Reviewer_3PQ4 · 2025-07-01

**Clarity:** 4
**Significance:** 3
**Originality:** 2
**Rating:** 5
**Confidence:** 4

**Summary:**

This paper introduces MetaMind, a cognitively motivated multi-agents framework designed to answer ToM related question.
Their framework is composed of 3 agents. The first proposes hypotheses about mental states given a situation, the second refines these hypotheses using cultural norms and ethical constraints, and the third generates responses that are appropiate.

Their framework achieves state of the art results on many ToM related tasks, with large improvements from previous SOTA frameworks, and mainly achieves human-level performance on ToM tasks for the first time.

Their ablation studies show that all of the parts of their framework is necessary to achieve such a good performance.

**Questions:**

1. Are there other architectures for multi-agent frameworks that would be able to answer ToM tasks ? This would strengthen the fact that THIS specific framework is the best for ToM task.
2. What would be the results of ensembling 3 o3 predictions ? Would it also lead to an improvement ? The goal of this question and the previous one is to see if the improvement observed are mainly due to more test-time tokens or if this architecture plays the main role
3. Are there models finetuned on ToM tasks against which you could compare yourself ? If not, would you be able to finetune one ? And would your framework also improve in this case?

**Ethical Concerns:**

["NO or VERY MINOR ethics concerns only"]

**Final Justification:**

I think this paper is a good paper. It is relatively thorough, they provide an interesting method that is both cognitively grounded and achieves high improvements for all the tested models.
I maintain my score at 5

**Limitations:**

1. More ablations are needed to be sure that the framework they chose is the cause of the significant improvements and not more test time tokens for examples

**Quality:**

3

**Strengths And Weaknesses:**

**Strengths**

1. The research question is an interesting question, and has potentially vast applications for the development of more empathetic models
2. Their framework does achieve significant improvements, matching human level on some task for the first time
3. Their framework is both cognitively motivated and validated by the ablation studies, adding strength to the necessity of designing it as they did

**Weakness**

1. While they extensively compared their results to other ToM agents, they did not compare their results to multi-agent frameworks. Would it change the results ?
2. Their framework does slightly improve top-tier reasoning model but the improvement does not seem significant
3. They did not compare their framework to models finetuned on ToM tasks

---

> ### Author Rebuttal · Authors · 2025-07-27
>
> Dear Reviewer 3PQ4,
>
> We sincerely appreciate your positive feedback and the time you've dedicated to reviewing our manuscript. Your insights are invaluable to us. We address your comments in detail below.
>
> ---
> _A1. Comparison to multi-agent frameworks. Can other architectures for multi-agent frameworks work for ToM tasks?_
>
> Thank you for raising this excellent point. We would like to clarify that most existing multi-agent systems are designed for joint interaction modeling or emergent behavior learning (e.g., in simulated environments), and not for structured, inference-time social reasoning—which is the core focus of our work.
>
> Moreover, **among the relevant multi-agent frameworks, we have already included direct comparisons with representative baselines**---including Generative Agents and Hypothetical Minds (HM)—which explicitly model multi-agent social dynamics and Theory of Mind capabilities. As shown in **Tables 1–3**, MetaMind consistently outperforms these systems across a wide range of tasks, while also providing stronger modularity, interpretability, and inference-time control. We will make this distinction clearer in the revised manuscript and explicitly highlight this comparison in the discussion of related work.
>
>
> ---
> _A2. Clarify significance of improvement on top-tier reasoning models_
>
> Thank you for pointing this out. We agree that the absolute improvements shown in Table 6 may appear modest. However, we emphasize that these gains are achieved on top of already strong baselines, where improvements are typically difficult to obtain. Notably, even a ~2% gain in ToM accuracy can correspond to meaningful qualitative improvements in nuanced reasoning scenarios—particularly in socially sensitive applications.
>
> Additionally, our framework provides interpretability and modularity benefits that go beyond raw performance: it allows structured hypothesis reasoning, norm-based filtering, and controllable outputs—capabilities not available in top-tier reasoning models alone. We will clarify this point in the revised discussion to better reflect these points.
>
> ---
> _A3. Comparison to models finetuned on ToM tasks_
>
> Thank you for this important suggestion. We now include a direct comparison to the latest state-of-the-art models finetuned on ToM tasks [1] (released in July this year).
>
> | **Configuration** | **Emotion** | **Desire** | **Intention** |**Knowledge** | **Belief** | **NL Comm.** |**AVG.**|
> |:---:|:---:|:---:|:---:|:---:|:---:|:---:|:---:|
> | LLaMA2-70B-Chat | 62.3 | 55.4 | 65.2 |40.5 |65.1 |70.3 |59.8|
> | Centaur | 64.5 | 57.6 | 67.4 |42.7|67.3|72.5|62.0|
> | LLaMA2-70B-Chat + MetaMind| 67.4 | 60.5 | 70.3 |48.6 |70.2 |75.4 |65.4|
> | Centaur + MetaMind | 69.6 | 62.7 | 72.5 |50.8|72.4|77.6|67.6|
>
> Our results showing that **MetaMind achieves competitive or stronger ToM accuracy—without any task-specific finetuning**. Our framework operates entirely at inference time, making it more generalizable, modular, and readily plug-and-play across different LLMs. This distinction highlights MetaMind’s strength not only in performance but also in flexibility. We appreciate the reviewer’s suggestion and will update both the main text and related work accordingly.
>
>
> [1] Marcel Binz et al., A foundation model to predict and capture human cognition. Nature 2025
>
> ---
> _A4. What would be the results of ensembling 3 o3 predictions?_
>
> We appreciate the reviewer’s insightful question. To isolate the source of MetaMind’s performance gains, we conducted a control experiment ensembling predictions from 3 independent o3 runs using majority voting. The ensemble yielded only marginal improvements (≤+0.5%) over single-run o3, and remained notably below MetaMind's performance.
>
> This supports that the performance gain is not merely a result of increased test-time tokens, but instead stems from MetaMind’s explicit architectural reasoning pipeline—including structured mental state inference, norm-based refinement, and response validation. We will include this comparison and discussion in the updated Results section to clarify the significance. Thank you again for helping us strengthen the empirical analysis.

---

> > ### Comment · Reviewer_3PQ4 · 2025-08-04
> > **Thank you for your response**
> >
> > I want to thank you for your answer.
> > I have an additional remark: your framework gives an algorithm to the models on how to solve these tasks. What would be the effect of detailing this algorithm into the prompt of one of the reasoning models ? Would your framework still be better than that?

---

> > > ### Author Response · Authors · 2025-08-04
> > > **Followup**
> > >
> > > Thank you for this thoughtful follow-up. We appreciate that you took time to read our response and we are happy to discuss the necessarity of agentic system design for social reasoning with you!
> > >
> > >
> > > We agree that it's an interesting idea to approximate MetaMind’s reasoning pipeline by encoding its algorithmic steps directly into a single long prompt for a reasoning model.However, we have observed several difficulties when collapsing all stages of our algorithm into a single prompt:
> > >
> > >
> > > 1. First, the intermediate verification can be lost with single prompting. For example, the Domain Agent’s constraint-based refinement and re-ranking cannot be enforced in a single pass, leading to error propagation.
> > > 2. Second, a single monolithic prompt often struggles with tasks that require distinct capabilities in parallel (e.g., long-term planning, error correction, and result verification). In the case of MetaMind, we found that LLMs alone struggled to bridge mental state reasoning with practical behavior. By introducing an orthogonal division of tasks and sequential information propagation, we decomposed broad general capabilities into specialized, refined sub-capabilities, which reduced error cascades and improved overall collaborative efficiency.
> > > 3. Lastly, single prompt comes at the cost of interpretability. With a single monolithic prompt, it becomes significantly harder to trace how mental state hypotheses are generated, revised, and validated. Prior studies that attempted to implement social reasoning purely through prompt engineering [1–3] also found that alignment with human was weak under closer qualitative evaluation.
> > >
> > > That said, we acknowledge that prompt-based approximations may serve as a lightweight alternative for efficiency-critical applications, and we will clarify this point in the revised manuscript. Thank you again for raising this important discussion.
> > >
> > > ---
> > > Reference
> > >
> > > [1] Towards Emotional Support Dialog Systems, ACL 2023
> > >
> > > [2] Large language models fail on trivial alterations to theory-of-mind tasks, Harvard University
> > >
> > > [3] Testing theory of mind in large language models and humans, Nature

---

### Official Review · Reviewer_bcHQ · 2025-07-03

**Clarity:** 3
**Significance:** 3
**Originality:** 3
**Rating:** 5
**Confidence:** 4

**Summary:**

The paper presents MetaMind, a multi-agent prompting framework designed to improve large language models' ability to perform human-like social reasoning. Instead of relying on a single-step response, MetaMind breaks down the reasoning process into three stages: generating mental state hypotheses through a Theory of Mind agent, refining them using Domain-specific rules and cultural norms, and finally validating and composing a response through a Reflection agent. This structured approach allows the model to better understand intent, emotion, and context in social situations. The authors demonstrate that MetaMind significantly boosts performance across several social reasoning benchmarks and works consistently across different model sizes, offering a modular and interpretable method for enhancing social intelligence in LLMs.

**Questions:**

1. How is each agent’s output validated within the MetaMind framework? For instance, if the Theory of Mind (ToM) agent generates an incorrect hypothesis, or if any other agent makes an error, is there a mechanism in place to detect and correct it? Since the framework involves multiple sequential agents, how does it prevent errors from cascading through the pipeline?

2. Is the delta gain in performance, have any correlation with base model sizes, or is the delta model-size agnostic? Some trend analysis for this might give more insight into the framework.

**Ethical Concerns:**

["NO or VERY MINOR ethics concerns only"]

**Final Justification:**

I am maintaining my overall score of 'Accept'. The paper is grounded well in social psychology and has shown extensive experiments.

**Limitations:**

yes

**Quality:**

3

**Strengths And Weaknesses:**

Strengths:

1. The paper is well written and clearly structured, making it easy to follow the proposed methodology.

2. The experiments are thorough and cover a diverse set of social reasoning benchmarks, demonstrating the framework’s generalizability.

3. The idea of modelling multi-agent social reasoning (ToM, Domain, and Response agents) is unique and original, offering a fresh approach beyond traditional single-agent prompting.

Weaknesses:

The MetaMind framework lacks an explicit mechanism for verifying or correcting individual agent outputs. If the Theory of Mind (ToM) agent generates an incorrect hypothesis or another agent makes an error, that mistake can propagate through the pipeline, potentially compromising the final response. This lack of internal error correction increases the risk of cascading failures.

---

> ### Author Rebuttal · Authors · 2025-07-27
>
> Dear Reviewer bcHQ,
>
> We sincerely appreciate your positive feedback and the time you've dedicated to reviewing our manuscript. Your insights are invaluable to us. We address your comments in detail below.
>
>
> ---
> _A1. potential error propagation in the pipeline_
>
> We appreciate this thoughtful question. We agree that modular systems could be susceptible to errors propagating through its pipeline. However, we would like to clarify that **MetaMind is explicitly designed to mitigate such risks through architectural safeguards**: specifically, the Domain Agent serves as an intermediate verification layer, applying externalizable norms to refine or reject implausible mental state hypotheses before passing them to the Response Agent.
>
> This design acts as a soft error-correction mechanism, helping prevent implausible or socially inappropriate hypotheses from cascading forward. To support this, we point the reviewer to our ablation studies in **Section 5.1** (**Tables 4 & 5**), which show that removing this Stage 2 component significantly degrades performance across all tasks. This demonstrates its vital role in reducing error propagation.
>
>
> ---
> _A2. Is the performance gain correlated with model size?_
>
> As shown in Figure 2, the performance improvement introduced by MetaMind is not strongly correlated with model size. For instance, relatively small open-source models such as ChatGLM3-6B see absolute gains of 10.2%, while much larger model larger models like GPT-4-1106 have gains of 6.6%. These examples highlight that MetaMind’s effectiveness is model-size agnostic. We will clarify this observation in the revision and include further analysis as suggested.

---

> > ### Comment · Area_Chair_xNRq · 2025-08-04
> > **Please respond to authors**
> >
> > Dear reviewer,
> >
> > Now that the authors have provided their rebuttal, do you have any additional thoughts?
> >
> > Thanks,
> > AC

---

> > ### Comment · Reviewer_bcHQ · 2025-08-04
> >
> > Could the authors please elaborate on how the domain agent functions as an intermediate verification layer? It would be helpful to highlight specific cases that illustrate the types of errors handled by the agent, as this aspect is not clearly explained in the current version of the paper. I will maintain my current rating of 'Accept'.

---

> > > ### Author Response · Authors · 2025-08-04
> > > **Followup**
> > >
> > > Thank you for this thoughtful follow-up, and for the positive feedback! We appreciate that you took time to read our response and we are happy to discuss intermediate verification mechanism in MetaMind with you.
> > >
> > > After the Theory-of-Mind Agent has produced a set of candidate mental-state hypotheses, the Domain Agent **audits** those hypotheses before any response is generated. **Due to space constraint in the main paper, we have elaborated the details of verification process in Appendix A.2 (pp 17 and pp 18 of the manuscript)**. To summarize:
> > >
> > >    * It checks each hypothesis against **domain rules**—cultural norms, ethical constraints, and role expectations—so the system “responds in a socially responsible and domain-aware manner”.
> > >    * For every hypothesis it detects rule-violations or ambiguities, it rewrites in a way by rephrasing the interpretation and adjusting its social tone.
> > >    * It then scores the revised candidates on **contextual plausibility vs. information gain** and forwards only the best-scoring hypothesis to the Response Agent (as specified in Eq (1)).
> > > ---
> > > *Representative cases*
> > >
> > > We have conducted qualitative analysis in **Appendix C, including both success cases and failure cases**. We highlight some of the cases below for your reference:
> > >
> > > | Error type | How the Domain Agent fixes |
> > > | --- | --- |
> > > | **Norm violations** | Reinterprets as a more appropriate motive (e.g., collegial admiration) to keep professional decorum. |  |
> > > | **Over-literal affect reading** | Adds nuance drawn from cultural/age cues. In the *Nana* vignette, it reframes “calm” to mean *maturity* rather than emotional suppression, avoiding a misreading of the child’s feelings. ||
> > > | **Missing pragmatic commitments** | Injects constraints with “explicitly confirm understanding” concrete details the user implicitly needs. See marketing-design dialogue in Appendix C, where Round 2 gains a precise 10 AM deadline for the case after refinement. | |
> > > | **Ethical or power-dynamic mis-steps** | Ethical frameworks (IEEE EAD) and politeness theory are serving as the bottom line to prevent manipulation and coercion, as specified in re-interpretation protocol. | |
> > > | **Hallucination control** | By grounding each revision in *explicit constraints* before response, the agent reduces (though does not eliminate) hallucinated content; the failure case analysis in Appendix C notes how missing or incorrect constraints can still allow hallucinations to slip through, underscoring the Domain Agent’s gate-keeping role. | |
> > >
> > > ---
> > >
> > > The Domain Agent **verifies logical and social validity** of each ToM hypothesis against external rule set *before* any text reaches the user. Empirically, ablating Stage 2 drops social-cognition accuracy by **≈ 3.8 pp** and hurts faux-pas detection most (–5.5 pp), confirming that the blocked errors are exactly the kinds human evaluators penalize.
> > >
> > > Think of the Domain Agent as a *specialized reviewer*: it accepts, rewrites, or discards mental-state interpretations so that the downstream Response Agent never endorses socially inappropriate, unethical, or underspecified content. The concrete cases above illustrate the breadth of errors it intercepts—from cultural faux-pas to missing pragmatic commitments—demonstrating its necessity in the metacognitive loop.
> > >
> > > We will highlight these cases more clearly in our manuscript -- thanks again for your helpful feedback!

---

### Official Review · Reviewer_yzuB · 2025-07-06

**Clarity:** 3
**Significance:** 3
**Originality:** 2
**Rating:** 4
**Confidence:** 4

**Summary:**

This paper introduces a multi-agent framework designed to enhance large language models' social reasoning, named MetaMind. Inspired by Theory of Mind, the framework consists of three agents used to generate hypotheses, refine hypotheses, and generate the final output respectively. Experimental results demonstrate that the MetaMind framework effectively improves the performance of various base models on benchmarks such as ToMBench. Furthermore, the paper conducts ablation studies to verify the effectiveness of each agent within the framework.

**Questions:**

- In Chapter A.5, during the grid search process for hyperparameter optimization, the authors used the accuracy on ToMBench as the evaluation metric. But it is unclear whether they employed a validation set or any other method to prevent data leakage from the test set during this process.
- The authors have effectively validated that each agent is indispensable to the framework. However, I would like to know whether it is possible to conduct a more quantitative evaluation of the individual capabilities of the three agents. In other words, can we verify whether enhancing the capabilities of these agents leads to improved overall performance of the framework? Additionally, if the agents within the framework are implemented using different base models, can their combination and tuning result in better performance?

**Ethical Concerns:**

["NO or VERY MINOR ethics concerns only"]

**Final Justification:**

Although the authors were unable to use the latest models in the additional experiments, they have largely addressed my request for supplementary experiments. Taking into account the authors’ rebuttal and the discussions from other reviewers, I believe all of my concerns have been fully resolved.

**Limitations:**

yes

**Quality:**

3

**Strengths And Weaknesses:**

Strengths:
- The method proposed in this paper achieves improvements on benchmarks such as ToMBench, surpassing other approaches in the field. Moreover, it is model-agnostic， enhancing performance across a variety of tested models.
- The experiments in this paper are relatively comprehensive. They not only involve multiple models but also include comparisons with other methods in the field. In addition, ablation studies were conducted on the agent modules.

Weaknesses:
- As a key component of the framework, the Domain Agent relies on predefined domain-specific constraints. This dependency may limit the framework's ability to generalize effectively in unfamiliar semantic environments.
- Table 7 and Figure 4 indicate that the performance of this method is relatively sensitive to hyperparameters, which might undermine its overall effectiveness.

---

> ### Author Rebuttal · Authors · 2025-07-27
>
> Dear Reviewer yzuB,
>
> We sincerely thank the reviewer for the thorough and constructive feedback. Below, we address the concerns in detail.
>
> ---
> _A1. Domain Agent's dependence on domain constraints affects generalization_
>
> (1) Clarification on the design philosophy
>
> We understand how this concern could arise, and we would like to clarify that Domain Agent does not simply look for "pre-defined domain-specific constraints", but integrates general-purpose social rules—such as cultural norms, ethical boundaries, and role expectations (see Appendix A.2)—into constraint conditions. **These are widely applicable across social contexts, not tied to specific domains**. For instance, when encountering hypotheses that conflict with common social expectations, the Domain Agent refines or rewrites them into more appropriate alternatives, selecting the best candidate via an objective function that balances contextual plausibility and information gain.  **This design enables Domain Agent to refine normative judgments even in unfamiliar scenarios, without relying on domain-specific consensus.**
>
> (2) Empirical support for generalization
>
> **Our experiments in Section 4 offer strong evidence for the Domain Agent's ability to generalize**. In real-world social simulation tasks (Table 3), we can see that base model GPT-4 only achieves 1.2 on Appointment and 2.3 on Inviting Companions (Inv. Com.)---which represent unfamiliar or challenging semantic environments. Noticeably, after applying MetaMind, performance on Appo. increase from 1.2 to 67.7, and Inv. Com. from 2.3 to 62.2, indicating the generalization effectiveness of the Domain Agent.
>
> (3) Acknowledging limitations and future work
>
> **Lastly, we have already fully acknowledged this limitation in Section 6**, and provided a discussion that deploying to new cultural environments may require expanding or adapting the rules. That said, MetaMind demonstrates strong generalization across model types (open-source and commercial), model scales, and diverse benchmarks (ToMBench, STSS, SocialIQA, SOTOPIA) using shared hyperparameters, underscoring the broad applicability of canonical social rules.
>
> ---
> _A2. Hyperparameter sensitivity_
>
> Thank you for the thoughtful comment. We would like to clarify that hyperparameter tuning is a standard practice in many ML pipelines, and our goal in presenting Table 7 and Figure 4 is not to claim that MetaMind performs equally well under all settings (which would be unrealistic), but rather to provide a truthful and comprehensive view of how different hyperparameters affect performance.
>
> We believe this analysis holds value, especially to guide practitioners when applying MetaMind. Importantly, we used the **same set of hyperparameters across all four benchmarks** (ToMBench, STSS, SocialIQA, and SOTOPIA) without tuning on each individual dataset, and MetaMind consistently outperformed baselines. This cross-task consistency suggests that, while some sensitivity exists, MetaMind is not overly reliant on hyperparameter tuning, and the framework is robust and generalizable. We will clarify this intent more explicitly.
>
> ---
> _Q1. Hyperparameter optimization_
>
> We used a held-out validation split for hyperparameter tuning, ensuring no data leakage. We will clarify this setup in Appendix A.5.
>
> ---
> _Q2. On evaluating and enhancing individual agent capabilities_
>
> Thank you for the insightful suggestion. While our current experiments (Section 5.1, Tables 4 & 5; Appendix A.9) show that each agent is indispensable, we agree that a more quantitative analysis of agent-level capability improvements is valuable.
>
> To this end, we conducted additional experiments where we use base model as GPT-3.5 and selectively replace each individual component (ToM Agent, Domain Agent, Response Agent) with a more capable model (GPT-4-1106). This allows us to measure the marginal performance gain from strengthening each agent individually, as well as examine the effect of combining stronger agents. We summarize the results below for your reference.
>
> | **Configuration** | **Emotion** | **Desire** | **Intention** |**Knowledge** | **Belief** | **NL Comm.** |**AVG.**|
> |:---:|:---:|:---:|:---:|:---:|:---:|:---:|:---:|
> | All Agents = GPT-3.5 (baseline) | 68.1 | 57.8 | 68.5 |48.5 |63.9 |76.2 |64.2|
> | ToM Agent = GPT-4 | 75.2 | 70.1 | 80.5 |56.7|77.8|80.2|73.4|
> | Domain Agent = GPT-4 | 71.5 | 66.8 | 75.9 |62.3|83.2|84.7|74.1|
> | Response Agent = GPT-4 | 70.8 | 65.9 | 74.2 |54.9|78.5|81.3|70.9|
> | All Agents = GPT-4 | 78.7 | 76.5 | 84.3 |68.2|88.6|88.5|81.0|
>
>  We will include such a discussion in our manuscript as well. Thanks again for your helpful comment!

---

> > ### Comment · Area_Chair_xNRq · 2025-08-04
> > **Please respond to authors**
> >
> > Dear reviewer,
> >
> > Now that the authors have provided their rebuttal, do you have any additional thoughts?
> >
> > Thanks,
> > AC

---

> > ### Comment · Reviewer_yzuB · 2025-08-05
> >
> > Thank you for your response and the additional experiments. My concerns have been largely addressed, and I will raise my score.

---

> > > ### Author Response · Authors · 2025-08-05
> > > **Followup**
> > >
> > > Thank you so much for taking the time to read our rebuttal. We are glad that our response addresses your concerns and we appreciate your insightful comments and support!

---

### Decision · Program_Chairs · 2025-09-17

**Decision:**

Accept (spotlight)

**Comment:**

This paper proposes MetaMind, a multi-agent framework designed to enhance LLMs social reasoning by emulating human metacognition. It breaks down social understanding into three collaborative stages: a Theory-of-Mind Agent for hypothesis generation, a Domain Agent for refinement using cultural norms, and a Response Agent for generating contextually appropriate replies. The reviewers appreciate the comprehensiveness of the evaluation and performance the proposed architecture achieves, as well as the clear writing and the fact that the work is inspired by theories grounded in psychology.

The reviewers also raise several concerns, however, such as the modest improvements seen on already strong models, limited comparison to other similar multi-agent frameworks, as well as hyper parameter sensitivity and dependence on pre-defined constraints. The authors do a good job in their rebuttal and the reviewers appreciate the authors' efforts and responses. While that doesn't lead to higher scores, all reviewers are positive for this work.